# AN EXPONENTIAL LEARNING RATE SCHEDULE FOR DEEP LEARNING

**Zhiyuan Li**
Princeton University
zhiyuanli@cs.princeton.edu

**Sanjeev Arora**
Princeton University and Institute for Advanced Study
arora@cs.princeton.edu

## ABSTRACT

Intriguing empirical evidence exists that deep learning can work well with exotic schedules for varying the learning rate. This paper suggests that the phenomenon may be due to Batch Normalization or BN(Ioffe & Szegedy, 2015), which is ubiquitous and provides benefits in optimization and generalization across all standard architectures. The following new results are shown about BN with weight decay and momentum (in other words, the typical use case which was not considered in earlier theoretical analyses of stand-alone BN (Ioffe & Szegedy, 2015; Santurkar et al., 2018; Arora et al., 2018)

- Training can be done using SGD with momentum and an exponentially *increasing* learning rate schedule, i.e., learning rate increases by some $(1 + \alpha)$ factor in every epoch for some $\alpha > 0$. (Precise statement in the paper.) To the best of our knowledge this is the first time such a rate schedule has been successfully used, let alone for highly successful architectures. As expected, such training rapidly blows up network weights, but the network stays well-behaved due to normalization.

- Mathematical explanation of the success of the above rate schedule: a rigorous proof that it is *equivalent* to the standard setting of BN + SGD + Standard Rate Tuning + Weight Decay + Momentum. This equivalence holds for other normalization layers as well, Group Normalization(Wu & He, 2018), Layer Normalization(Ba et al., 2016), Instance Norm(Ulyanov et al., 2016), etc.

- A worked-out toy example illustrating the above linkage of hyper-parameters. Using either weight decay or BN alone reaches global minimum, but convergence fails when both are used.

## 1 INTRODUCTION

Batch Normalization (BN) offers significant benefits in optimization and generalization across architectures, and has become ubiquitous. Usually best performance is attained by adding weight decay and momentum in addition to BN.

Usually weight decay is thought to improve generalization by controlling the norm of the parameters. However, it is fallacious to try to separately think of optimization and generalization because we are dealing with a nonconvex objective with multiple optima. Even slight changes to the training surely lead to a different trajectory in the loss landscape, potentially ending up at a different solution! One needs trajectory analysis to have a hope of reasoning about the effects of such changes.

In the presence of BN and other normalization schemes, including GroupNorm, LayerNorm, and InstanceNorm, the optimization objective is *scale invariant* to the parameters, which means rescaling parameters would not change the prediction, except the parameters that compute the output which do not have BN. However, Hoffer et al. (2018b) shows that fixing the output layer randomly doesn't harm the performance of the network. So the trainable parameters satisfy scale invariance.(See more in Appendix E) The current paper introduces new modes of analysis for such settings. This rigorous analysis yields the surprising conclusion that the original learning rate (LR) schedule and weight decay(WD) can be folded into a new exponential schedule for learning rate: in each iteration multiplying it by $(1 + \alpha)$ for some $\alpha > 0$ that depends upon the momentum and weight decay rate.

**Theorem 1.1** (Main, Informal). SGD on a scale-invariant objective with initial learning rate $\eta$, weight decay factor $\lambda$, and momentum factor $\gamma$ is equivalent to SGD with momentum factor $\gamma$ where at iteration $t$, the learning rate $\tilde{\eta}_t$ in the new exponential learning rate schedule is defined as $\tilde{\eta}_t = \alpha^{-2t-1}\eta$ without weight decay($\tilde{\lambda} = 0$) where $\alpha$ is a non-zero root of equation

$$x^2 - (1 + \gamma - \lambda\eta)x + \gamma = 0, \tag{1}$$

Specifically, when momentum $\gamma = 0$, the above schedule can be simplified as $\tilde{\eta}_t = (1 - \lambda\eta)^{-2t-1}\eta$.

The above theorem requires that the product of learning rate and weight decay factor, $\lambda\eta$, is small than $(1 - \sqrt{\gamma})^2$, which is almost always satisfied in practice. The rigorous and most general version of above theorem is Theorem 2.12, which deals with multi-phase LR schedule, momentum and weight decay.

There are other recently discovered exotic LR schedules, e.g. Triangular LR schedule(Smith, 2017) and Cosine LR schedule(Loshchilov & Hutter, 2016), and our exponential LR schedule is an extreme example of LR schedules that become possible in presence of BN. Such an exponential increase in learning rate seems absurd at first sight and to the best of our knowledge, no deep learning success has been reported using such an idea before. It does highlight the above-mentioned viewpoint that in deep learning, optimization and regularization are not easily separated. Of course, the exponent trumps the effect of initial lr very fast (See Figure 4), which explains why training with BN and WD is not sensitive to the scale of initialization, since with BN, tuning the scale of initialization is equivalent to tuning the initial LR $\eta$ while fixing the product of LR and WD, $\eta\lambda$ (See Lemma 2.7).

Note that it is customary in BN to switch to a lower LR upon reaching a plateau in the validation loss. According to the analysis in the above theorem, this corresponds to an exponential growth with a smaller exponent, except for a transient effect when a correction term is needed for the two processes to be equivalent (see discussion around Theorem 2.12).

Thus the final training algorithm is roughly as follows: *Start from a convenient LR like* 0.1*, and grow it at an exponential rate with a suitable exponent. When validation loss plateaus, switch to an exponential growth of LR with a lower exponent. Repeat the procedure until the training loss saturates.*

In Section 3, we demonstrate on a toy example how weight decay and normalization are inseparably involved in the optimization process. With either weight decay or normalization alone, SGD will achieve zero training error. But with both turned on, SGD fails to converge to global minimum.

In Section 4, we experimentally verify our theoretical findings on CNNs and ResNets. We also construct better exponential LR schedules by incorporating the Cosine LR schedule on CIFAR10, which opens the possibility of even more general theory of rate schedule tuning towards better performance.

## 1.1 RELATED WORK

There have been other theoretical analyses of training models with scale-invariance. (Cho & Lee, 2017) proposed to run Riemanian gradient descent on Grassmann manifold $\mathcal{G}(1, n)$ since the weight matrix is scaling invariant to the loss function. observed that the effective stepsize is proportional to $\frac{\eta_w}{\|\boldsymbol{w}_t\|^2}$. (Arora et al., 2019) show the gradient is always perpendicular to the current parameter vector which has the effect that norm of each scale invariant parameter group increases monotonically, which has an auto-tuning effect. (Wu et al., 2018) proposes a new adaptive learning rate schedule motivated by scale-invariance property of Weight Normalization.

**Previous work for understanding Batch Normalization.** (Santurkar et al., 2018) suggested that the success of BNhas does not derive from reduction in Internal Covariate Shift, but by making landscape smoother. (Kohler et al., 2018) essentially shows linear model with BN could achieve exponential convergence rate assuming gaussian inputs, but their analysis is for a variant of GD with an inner optimization loop rather than GD itself. (Bjorck et al., 2018) observe that the higher learning rates enabled by BN empirically improves generalization. (Arora et al., 2019) prove that with certain mild assumption, (S)GD with BN finds approximate first order stationary point with any fixed learning rate. None of the above analyses incorporated weight decay, but (Zhang et al., 2019; Hoffer et al., 2018a; van Laarhoven, 2017) argued qualitatively that weight decay makes parameters

have smaller norms, and thus the effective learning rate, $\frac{\eta_w}{\|\boldsymbol{w}_t\|^2}$ is larger. They described experiments showing this effect but didn't have a closed form theoretical analysis like ours. None of the above analyses deals with momentum rigorously.

## 1.2 PRELIMINARIES AND NOTATIONS

For batch $\mathcal{B} = \{x_i\}_{i=1}^B$, network parameter $\boldsymbol{\theta}$, we denote the network by $f_{\boldsymbol{\theta}}$ and the loss function at iteration $t$ by $L_t(f_{\boldsymbol{\theta}}) = L(f_{\boldsymbol{\theta}}, \mathcal{B}_t)$. When there's no ambiguity, we also use $L_t(\boldsymbol{\theta})$ for convenience.

We say a loss function $L(\boldsymbol{\theta})$ is *scale invariant* to its parameter $\boldsymbol{\theta}$ is for any $c \in \mathbb{R}^+$, $L(\boldsymbol{\theta}) = L(c\boldsymbol{\theta})$. In practice, the source of scale invariance is usually different types of normalization layers, including Batch Normalization (Ioffe & Szegedy, 2015), Group Normalization (Wu & He, 2018), Layer Normalization (Ba et al., 2016), Instance Norm (Ulyanov et al., 2016), etc.

Implementations of SGD with Momentum/Nesterov comes with subtle variations in literature. We adopt the variant from Sutskever et al. (2013), also the default in PyTorch (Paszke et al., 2017). $L2$ *regularization* (a.k.a. *Weight Decay*) is another common trick used in deep learning. Combining them together, we get the one of the mostly used optimization algorithms below.

**Definition 1.2.** [SGD with Momentum and Weight Decay] At iteration $t$, with randomly sampled batch $\mathcal{B}_t$, update the parameters $\boldsymbol{\theta}_t$ and momentum $\boldsymbol{v}_t$ as following:

$$\boldsymbol{\theta}_t = \boldsymbol{\theta}_{t-1} - \eta_{t-1}\boldsymbol{v}_t \tag{2}$$

$$\boldsymbol{v}_t = \gamma \boldsymbol{v}_{t-1} + \nabla_{\boldsymbol{\theta}}\left(L_t(\boldsymbol{\theta}_{t-1}) + \frac{\lambda_{t-1}}{2}\|\boldsymbol{\theta}_{t-1}\|^2\right), \tag{3}$$

where $\eta_t$ is the learning rate at epoch $t$, $\gamma$ is the momentum coefficient, and $\lambda$ is the factor of weight decay. Usually, $\boldsymbol{v}_0$ is initialized to be $\boldsymbol{0}$.

For ease of analysis, we will use the following equivalent of Definition 1.2.

$$\frac{\boldsymbol{\theta}_t - \boldsymbol{\theta}_{t-1}}{\eta_{t-1}} = \gamma \frac{\boldsymbol{\theta}_{t-1} - \boldsymbol{\theta}_{t-2}}{\eta_{t-2}} - \nabla_{\boldsymbol{\theta}}\left((L(\boldsymbol{\theta}_{t-1}) + \frac{\lambda_{t-1}}{2}\|\boldsymbol{\theta}_{t-1}\|_2^2\right), \tag{4}$$

where $\eta_{-1}$ and $\boldsymbol{\theta}_{-1}$ must be chosen in a way such that $\boldsymbol{v}_0 = \frac{\boldsymbol{\theta}_0 - \boldsymbol{\theta}_{-1}}{\eta_{-1}}$ is satisfied, e.g. when $\boldsymbol{v}_0 = \boldsymbol{0}$, $\boldsymbol{\theta}_{-1} = \boldsymbol{\theta}_0$ and $\eta_{-1}$ could be arbitrary.

A key source of intuition is the following simple lemma about scale-invariant networks Arora et al. (2019). The first property ensures GD (with momentum) always increases the norm of the weight.(See Lemma C.1 in Appendix C) and the second property says that the gradients are smaller for parameteres with larger norm, thus stabilizing the trajectory from diverging to infinity.

**Lemma 1.3** (Scale Invariance). *If for any $c \in \mathbb{R}^+$, $L(\boldsymbol{\theta}) = L(c\boldsymbol{\theta})$, then*
*(1). $\langle \nabla_{\boldsymbol{\theta}}L, \boldsymbol{\theta}\rangle = 0$;*
*(2). $\nabla_{\boldsymbol{\theta}}L\big|_{\boldsymbol{\theta}=\boldsymbol{\theta}_0} = c\nabla_{\boldsymbol{\theta}}L\big|_{\boldsymbol{\theta}=c\boldsymbol{\theta}_0}$, for any $c > 0$*

## 2 DERIVING EXPONENTIAL LEARNING RATE SCHEDULE

As a warm-up in Section 2.1 we show that if momentum is turned off then *Fixed LR + Fixed WD* can be translated to an equivalent *Exponential LR*. In Section 2.2 we give a more general analysis on the equivalence between *Fixed LR + Fixed WD + Fixed Momentum Factor* and *Exponential LR + Fixed Momentum Factor*. While interesting, this still does completely apply to real-life deep learning where reaching full accuracy usually requires multiple phases in training where LR is fixed within a phase and reduced by some factor from one phase to the next. Section 2.3 shows how to interpret such a multi-phase LR schedule + WD + Momentum as a certain multi-phase exponential LR schedule with Momentum.

### 2.1 REPLACING WD BY EXPONENTIAL LR IN MOMENTUM-FREE SGD

We use notation of Section 1.2 and assume LR is fixed over iterations, i.e. $\eta_t = \eta_0$, and $\gamma$ (momentum factor) is set as 0. We also use $\lambda$ to denote WD factor and $\boldsymbol{\theta}_0$ to denote the initial parameters.

The intuition should be clear from Lemma 1.3, which says that shrinking parameter weights by factor $\rho$ (where $\rho < 1$) amounts to making the gradient $\rho^{-1}$ times larger without changing its direction. Thus in order to restore the ratio between original parameter and its update (LR×Gradient), the easiest way would be scaling LR by $\rho^2$. This suggests that scaling the parameter $\boldsymbol{\theta}$ by $\rho$ at each step is equivalent to scaling the LR $\eta$ by $\rho^{-2}$.

To prove this formally we use the following formalism. We'll refer to the vector $(\boldsymbol{\theta}, \eta)$ the *state* of a training algorithm and study how this evolves under various combinations of parameter changes. We will think of each step in training as a *mapping* from one state to another. Since mappings can be composed, any finite number of steps also correspond to a mapping. The following are some basic mappings used in the proof.

1. Run GD with WD for a step: $\quad \mathrm{GD}_t^\rho(\boldsymbol{\theta}, \eta) = (\rho\boldsymbol{\theta} - \eta\nabla L_t(\boldsymbol{\theta}), \eta);$
2. Scale the parameter $\boldsymbol{\theta}$: $\quad \Pi_1^c(\boldsymbol{\theta}, \eta) = (c\boldsymbol{\theta}, \eta);$
3. Scale the LR $\eta$: $\quad \Pi_2^c(\boldsymbol{\theta}, \eta) = (\boldsymbol{\theta}, c\eta).$

For example, when $\rho = 1$, $\mathrm{GD}_t^1$ is vanilla GD update without WD, also abbreviated as $\mathrm{GD}_t$. When $\rho = 1 - \lambda\eta_0$, $\mathrm{GD}_t^{1-\lambda\eta_0}$ is GD update with WD $\lambda$ and LR $\eta_0$. Here $L_t$ is the loss function at iteration $t$, which is decided by the batch of the training samples $\mathcal{B}_t$ in $t$th iteration. Below is the main result of this subsection, showing our claim that GD + WD $\Leftrightarrow$ GD+ Exp LR (when Momentum is zero). It will be proved after a series of lemmas.

**Theorem 2.1** (WD $\Leftrightarrow$ Exp LR). For every $\rho < 1$ and positive integer $t$ following holds:

$$\mathrm{GD}_{t-1}^\rho \circ \cdots \circ \mathrm{GD}_0^\rho = \left[\Pi_1^{\rho^t} \circ \Pi_2^{\rho^{2t}}\right] \circ \Pi_2^{\rho^{-1}} \circ \mathrm{GD}_{t-1} \circ \Pi_2^{\rho^{-2}} \circ \cdots \circ \mathrm{GD}_1 \circ \Pi_2^{\rho^{-2}} \circ \mathrm{GD}_0 \circ \Pi_2^{\rho^{-1}}.$$

With WD being $\lambda$, $\rho$ is set as $1 - \lambda\eta_0$ and thus the scaling factor of LR per iteration is $\rho^{-2} = (1 - \lambda\eta_0)^{-2}$, except for the first iteration it's $\rho^{-1} = (1 - \lambda\eta_0)^{-1}$.

We first show how to write GD update with WD as a composition of above defined basic maps.

**Lemma 2.2.** $GD_t^\rho = \Pi_2^\rho \circ \Pi_1^\rho \circ GD_t \circ \Pi_2^{\rho^{-1}}.$

Below we will define the proper notion of equivalence such that (1). $\Pi_1^\rho \sim \Pi_2^{\rho^{-2}}$, which implies $\mathrm{GD}_t^\rho \sim \Pi_2^{\rho^{-1}} \circ \mathrm{GD}_t \circ \Pi_2^{\rho^{-1}}$; (2) the equivalence is preserved under future GD updates.

We first extend the equivalence between weights (same direction) to that between states, with additional requirement that the ratio between the size of GD update and that of parameter are the same among all equivalent states, which yields the notion of *Equivalent Scaling*.

**Definition 2.3** (Equivalent States). $(\boldsymbol{\theta}, \eta)$ is *equivalent* to $(\boldsymbol{\theta}', \eta')$ iff $\exists c > 0$, $(\widetilde{\boldsymbol{\theta}}, \widetilde{\eta}) = [\Pi_1^c \circ \Pi_2^{c^2}](\boldsymbol{\theta}, \eta) = (c\boldsymbol{\theta}, c^2\eta)$, which is also denoted by $(\widetilde{\boldsymbol{\theta}}, \widetilde{\eta}) \overset{c}{\sim} (\boldsymbol{\theta}, \eta)$. $\Pi_1^c \circ \Pi_2^{c^2}$ is called *Equivalent Scaling* for all $c > 0$.

The following lemma shows that equivalent scaling commutes with GD update with WD, implying that equivalence is preserved under GD update (Lemma 2.4). This anchors the notion of equivalence — we could insert equivalent scaling anywhere in a sequence of basic maps(GD update, LR/parameter scaling), without changing the final network.

**Lemma 2.4.** *For any constant $c, \rho > 0$ and $t \geq 0$, $GD_t^\rho \circ [\Pi_1^c \circ \Pi_2^{c^2}] = [\Pi_1^c \circ \Pi_2^{c^2}] \circ GD_t^\rho$.* In other words, $(\boldsymbol{\theta}, \eta) \overset{c}{\sim} (\boldsymbol{\theta}', \eta') \implies GD_t^\rho(\boldsymbol{\theta}, \eta) \overset{c}{\sim} GD_t^\rho(\boldsymbol{\theta}', \eta')$.

Now we formally define equivalence relationship between maps using equivalent scalings.

**Definition 2.5** (Equivalent Maps). Two maps $F, G$ are *equivalent* iff $\exists c > 0$, $F = \Pi_1^c \circ \Pi_2^{c^2} \circ G$, which is also denoted by $F \overset{c}{\sim} G$.

*Proof of Theorem 2.1.* By Lemma 2.2., $\mathrm{GD}_t^\rho \overset{\rho}{\sim} \Pi_2^{\rho^{-1}} \circ \mathrm{GD}_t \circ \Pi_2^{\rho^{-1}}$. By Lemma 2.4, GD update preserves map equivalence, i.e. $F \overset{c}{\sim} G \Rightarrow \mathrm{GD}_t^\rho \circ F \overset{c}{\sim} \mathrm{GD}_t^\rho \circ G, \forall c, \rho > 0$. Thus,

$$\mathrm{GD}_{t-1}^\rho \circ \cdots \circ \mathrm{GD}_0^\rho \overset{\rho^t}{\sim} \Pi_2^{\rho^{-1}} \circ \mathrm{GD}_{t-1} \circ \Pi_2^{\rho^{-2}} \circ \cdots \circ \mathrm{GD}_1 \circ \Pi_2^{\rho^{-2}} \circ \mathrm{GD}_0 \circ \Pi_2^{\rho^{-1}}. \qquad \square$$

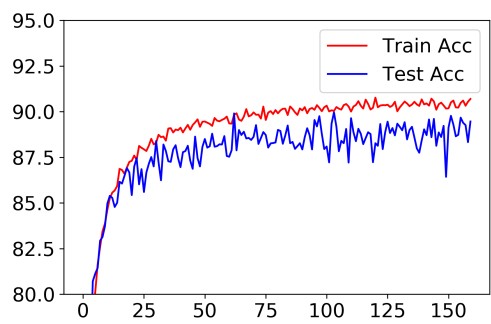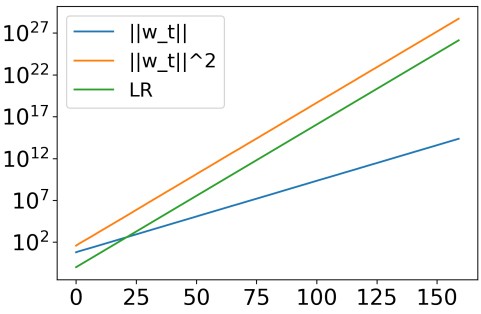

**Figure 1:** Taking PreResNet32 with standard hyperparameters and replacing WD during first phase (Fixed LR) by exponential LR according to Theorem 2.9 to the schedule $\widetilde{\eta}_t = 0.1 \times 1.481^t$, momentum 0.9. Plot on right shows weight norm $w$ of the first convolutional layer in the second residual block grows exponentially, satisfying $\frac{\|w_t\|^2}{\widetilde{\eta}_t}$ = constant. Reason being that according to the proof it is essentially the norm square of the weights when trained with Fixed LR + WD + Momentum, and published hyperparameters kept this norm roughly constant during training.

## 2.2 REPLACING WD BY EXPONENTIAL LR: CASE OF CONSTANT LR WITH MOMENTUM

In this subsection the setting is the same to that in Subsection 2.1 except that the momentum factor is $\gamma$ instead of 0. Suppose the initial momentum is $v_0$, we set $\theta_{-1} = \theta_0 - v_0\eta$. Presence of momentum requires representing the state of the algorithm with four coordinates, $(\theta, \eta, \theta', \eta')$, which stand respectively for the current parameters/LR and the buffered parameters/LR (from last iteration) respectively. Similarly, we define the following basic maps and equivalence relationships.

1. Run GD with WD for a step: $\quad \mathrm{GD}_t^\rho(\theta, \eta, \theta', \eta') = \left(\rho\theta + \eta\left(\gamma\frac{\theta-\theta'}{\eta'} - \nabla L_t(\theta)\right), \eta, \theta, \eta\right);$

2. Scale Current parameter $\theta$ $\quad \Pi_1^c(\theta, \eta, \theta', \eta') = (c\theta, \eta, \theta', \eta');$

3. Scale Current LR $\eta$: $\quad \Pi_2^c(\theta, \eta, \theta', \eta') = (\theta, c\eta, \theta', \eta');$

4. Scale Buffered parameter $\theta'$: $\quad \Pi_3^c(\theta, \eta, \theta', \eta') = (\theta, \eta, c\theta', \eta');$

5. Scale Buffered parameter $\eta'$: $\quad \Pi_4^c(\theta, \eta, \theta', \eta') = (\theta, \eta, \theta', c\eta').$

**Definition 2.6** (Equivalent States). $(\theta, \eta, \theta', \eta')$ is *equivalent* to $(\widetilde{\theta}, \widetilde{\eta}, \widetilde{\theta}', \widetilde{\eta}')$ iff $\exists c > 0$, $(\theta, \eta, \theta', \eta') = \left[\Pi_1^c \circ \Pi_2^{c^2} \circ \Pi_3^c \circ \Pi_4^{c^2}\right](\widetilde{\theta}, \widetilde{\eta}, \widetilde{\theta}', \widetilde{\eta}') = (c\widetilde{\theta}, c^2\widetilde{\eta}, c\widetilde{\theta}', c^2\widetilde{\eta}')$, which is also denoted by $(\theta, \eta, \theta', \eta') \overset{c}{\sim} (\widetilde{\theta}, \widetilde{\eta}, \widetilde{\theta}', \widetilde{\eta}')$. We call $\Pi_1^c \circ \Pi_2^{c^2} \circ \Pi_3^c \circ \Pi_4^{c^2}$ *Equivalent Scalings* for all $c > 0$.

Again by expanding the definition, we show equivalent scalings commute with GD update.

**Lemma 2.7.** $\forall c, \rho > 0$ and $t \geq 0$, $GD_t^\rho \circ \left[\Pi_1^c \circ \Pi_2^{c^2} \circ \Pi_3^c \circ \Pi_4^{c^2}\right] = \left[\Pi_1^c \circ \Pi_2^{c^2} \circ \Pi_3^c \circ \Pi_4^{c^2}\right] \circ GD_t^\rho$.

Similarly, we can rewrite $\mathrm{GD}_t^\rho$ as a composition of vanilla GD update and other scalings by expanding the definition, when the current and buffered LR are the same in the input of $\mathrm{GD}_t^\rho$.

**Lemma 2.8.** For any input $(\theta, \eta, \theta', \eta)$, if $\alpha > 0$ is a root of $\alpha + \gamma\alpha^{-1} = \rho + \gamma$, then
$$GD_t^\rho(\theta, \eta, \theta', \eta) = \left[\Pi_4^\alpha \circ \Pi_2^\alpha \circ \Pi_1^\alpha \circ GD_t \circ \Pi_2^{\alpha^{-1}} \circ \Pi_3^\alpha \circ \Pi_4^\alpha\right](\theta, \eta, \theta', \eta). \text{ In other words,}$$
$$GD_t^\rho(\theta, \eta, \theta', \eta) \overset{\alpha}{\sim} \left[\Pi_3^{\alpha^{-1}} \circ \Pi_4^{\alpha^{-1}} \circ \Pi_2^{\alpha^{-1}} \circ GD_t \circ \Pi_2^{\alpha^{-1}} \circ \Pi_3^\alpha \circ \Pi_4^\alpha\right](\theta, \eta, \theta', \eta). \quad (5)$$

Though looking complicated, the RHS of Equation 5 is actually the desired $\Pi_2^{\alpha^{-1}} \circ \mathrm{GD}_t \circ \Pi_2^{\alpha^{-1}}$ conjugated with some scaling on momentum part $\Pi_3^\alpha \circ \Pi_4^\alpha$, and $\Pi_3^{\alpha^{-1}} \circ \Pi_4^{\alpha^{-1}}$ in the current update cancels with the $\Pi_3^\alpha \circ \Pi_4^\alpha$ in the next update. Now we are ready to show the equivalence between WD and Exp LR schedule when momentum is turned on for both.

**Theorem 2.9** (GD + WD ⟺ GD+ Exp LR; With Momentum). The following defined two sequences of parameters $,\{\theta_t\}_{t=0}^\infty$ and $\{\widetilde{\theta}_t\}_{t=0}^\infty$, satisfy $\widetilde{\theta}_t = \alpha^t\theta_t$, thus they correspond to the same networks in function space, i.e. $f_{\theta_t} = f_{\widetilde{\theta}_t}$, $\forall t \in \mathbb{N}$, given $\widetilde{\theta}_0 = \theta_0$, $\widetilde{\theta}_{-1} = \theta_{-1}\alpha$, and $\widetilde{\eta}_t = \eta_0\alpha^{-2t-1}$.

1. $\frac{\theta_t - \theta_{t-1}}{\eta_0} = \frac{\gamma(\theta_{t-1}-\theta_{t-2})}{\eta_0} - \nabla_\theta(L(\theta_{t-1}) + \frac{\lambda}{2}\|\theta_{t-1}\|_2^2)$

2. $\frac{\widetilde{\theta}_t - \widetilde{\theta}_{t-1}}{\widetilde{\eta}_t} = \frac{\gamma(\widetilde{\theta}_{t-1}-\widetilde{\theta}_{t-2})}{\widetilde{\eta}_{t-1}} - \nabla_\theta L(\widetilde{\theta}_{t-1})$

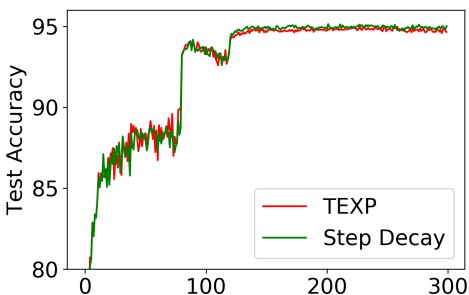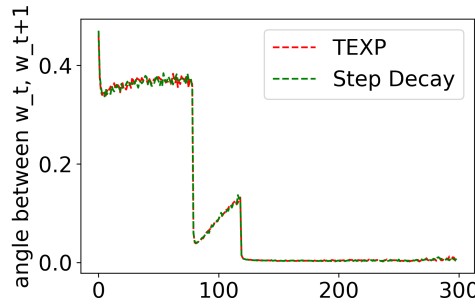

**Figure 2:** PreResNet32 trained with standard Step Decay and its corresponding Tapered-Exponential LR schedule. As predicted by Theorem 2.12, they have similar trajectories and performances.

where $\alpha$ is a positive root of equation $x^2 - (1 + \gamma - \lambda\eta_0)x + \gamma = 0$, which is always smaller than 1(See Appendix B.1). When $\gamma = 0$, $\alpha = 1 - \lambda\eta_0$ is the unique non-zero solution.

**Remark 2.10.** Above we implicitly assume that $\lambda\eta_0 \leq (1 - \sqrt{\gamma})^2$ such that the roots are real and this is always true in practice. For instance of standard hyper-parameters where $\gamma = 0.9, \eta_0 = 0.1, \lambda = 0.0005$, $\frac{\lambda\eta_0}{(1-\sqrt{\gamma})^2} \approx 0.019 \ll 1$.

*Proof.* Note that $(\widetilde{\boldsymbol{\theta}}_0, \widetilde{\eta}_0, \widetilde{\boldsymbol{\theta}}_{-1}, \widetilde{\eta}_{-1}) = \left[\Pi_2^{\alpha^{-1}} \circ \Pi_3^{\alpha} \circ \Pi_4^{\alpha}\right](\boldsymbol{\theta}_0, \eta_0, \boldsymbol{\theta}_0, \eta_0)$, it suffices to show that

$$\left[\Pi_3^{\alpha^{-1}} \circ \Pi_4^{\alpha^{-1}} \circ \Pi_2^{\alpha^{-1}} \circ \text{GD}_{t-1} \circ \Pi_2^{\alpha^{-2}} \circ \cdots \circ \text{GD}_1 \circ \Pi_2^{\alpha^{-2}} \circ \text{GD}_0 \circ \Pi_2^{\alpha^{-1}} \circ \Pi_3^{\alpha} \circ \Pi_4^{\alpha}\right](\boldsymbol{\theta}_0, \eta_0, \boldsymbol{\theta}_0, \eta_0)$$

$$\overset{\alpha^t}{\sim} \text{GD}_{t-1}^{1-\lambda\eta_0} \circ \cdots \circ \text{GD}_0^{1-\lambda\eta_0}(\boldsymbol{\theta}_0, \eta_0, \boldsymbol{\theta}_0, \eta_0), \quad \forall t \geq 0.$$

which follows immediately from Lemma 2.7 and Lemma 2.8 by induction. □

## 2.3 REPLACING WD BY EXPONENTIAL LR: CASE OF MULTIPLE LR PHASES

Usual practice in deep learning shows that reaching full training accuracy requires reducing the learning rate a few times.

**Definition 2.11.** *Step Decay* is the (standard) learning rate schedule, where training has $K$ phases $I = 0, 1, \ldots, K - 1$, where phase $I$ starts at iteration $T_I$ ($T_0 = 0$), and all iterations in phase $I$ use a fixed learning rate of $\eta_I^*$.

The algorithm state in Section 2.2, consists of 4 components including buffered and current LR. When LR changes, the buffered and current LR are not equal, and thus Lemma 2.8 cannot be applied any more. In this section we show how to fix this issue by adding extra momentum correction. In detail, we show the below defined Exp LR schedule leads the same trajectory of networks in function space, with one-time momentum correction at the start of each phase. We empirically find on CIFAR10 that ignoring the correction term does not change performance much.

**Theorem 2.12** (Tapered-Exponential LR Schedule). There exists a way to correct the momentum only at the first iteration of each phase, such that the following Tapered-Exponential LR schedule (TEXP) $\{\widetilde{\eta}_t\}$ with momentum factor $\gamma$ and no WD, leads the same sequence networks in function space as that of Step Decay LR schedule(Definition 2.11) with momentum factor $\gamma$ and WD $\lambda$.

$$\widetilde{\eta}_t = \begin{cases} \widetilde{\eta}_{t-1} \times (\alpha_{I-1}^*)^{-2} & \text{if } T_{I-1} + 1 \leq t \leq T_I - 1, I \geq 1; \\ \widetilde{\eta}_{t-1} \times \frac{\eta_I^*}{\eta_{I-1}^*} \times (\alpha_I^*)^{-1}(\alpha_{I-1}^*)^{-1} & \text{if } t = T_I, I \geq 1, \end{cases} \quad (6)$$

where $\alpha_I^* = \frac{1 + \gamma - \lambda\eta_I^* + \sqrt{(1+\gamma-\lambda\eta_I^*)^2 - 4\gamma}}{2}$, $\widetilde{\eta}_0 = \eta_0 \cdot (\alpha_0^*)^{-1} = \eta_0^* \cdot (\alpha_0^*)^{-1}$.

The analysis in previous subsection give the equivalence within each phase, where the same LR is used throughout the phase. To deal with the difference between buffered LR and current LR when entering new phases, the idea is to pretend $\eta_{t-1} = \eta_t$ and $\boldsymbol{\theta}_{t-1}$ becomes whatever it needs to maintain $\frac{\boldsymbol{\theta}_t - \boldsymbol{\theta}_{t-1}}{\eta_{t-1}}$ such that we can again apply Lemma 2.8, which requires the current LR of the input state is equal to its buffered LR. Because scaling $\alpha$ in RHS of Equation 5 is different in different phases, so unlike what happens within each phase, they don't cancel with each other at phase transitions, thus remaining as a correction of the momentum. The proofs are delayed to Appendix B, where we proves a more general statement allowing phase-dependent WD, $\{\lambda_I\}_{I=0}^{K-1}$.

**Alternative interpretation of Step Decay to exponential LR schedule:**Below we present a new LR schedule, *TEXP++*, which is exactly equivalent to *Step Decay* without the need of one-time correction of momentum when entering each phase. We further show in Appendix B.1 that when translating from *Step Decay*, the *TEXP++* we get is very close to the original *TEXP*(Equation 9), i.e. the ratio between the LR growth per round, $\frac{\widetilde{\eta}_{t+1}}{\widetilde{\eta}_t}/\frac{\widetilde{\eta}'_{t+1}}{\widetilde{\eta}'_t}$ converges to 1 exponentially each phase. For example, with WD 0.0005, max LR 0.1, momentum factor 0.9, the ratio is within $1 \pm 0.0015 *$ $0.9^{t-T_I}$, meaning TEXP and TEXP++ are very close for Step Decay with standard hyperparameters.

**Theorem 2.13.** The following two sequences of parameters ,$\{\boldsymbol{\theta}_t\}_{t=0}^{\infty}$ and $\{\widetilde{\boldsymbol{\theta}}_t\}_{t=0}^{\infty}$, define the same sequence of network functions, i.e. $f_{\boldsymbol{\theta}_t} = f_{\widetilde{\boldsymbol{\theta}}_t}$, $\forall t \in \mathbb{N}$, given the initial conditions, $\widetilde{\boldsymbol{\theta}}_0 = P_0 \boldsymbol{\theta}_0$, $\widetilde{\boldsymbol{\theta}}_{-1} = P_{-1} \boldsymbol{\theta}_{-1}$.

1. $\frac{\boldsymbol{\theta}_t - \boldsymbol{\theta}_{t-1}}{\eta_{t-1}} = \gamma \frac{\boldsymbol{\theta}_{t-1} - \boldsymbol{\theta}_{t-2}}{\eta_{t-2}} - \nabla_{\boldsymbol{\theta}} \left( (L(\boldsymbol{\theta}_{t-1}) + \frac{\lambda_{t-1}}{2} \|\boldsymbol{\theta}_{t-1}\|_2^2 \right)$, for $t = 1, 2, \ldots$;

2. $\frac{\widetilde{\boldsymbol{\theta}}_t - \widetilde{\boldsymbol{\theta}}_{t-1}}{\widetilde{\eta}_{t-1}} = \gamma \frac{\widetilde{\boldsymbol{\theta}}_{t-1} - \widetilde{\boldsymbol{\theta}}_{t-2}}{\widetilde{\eta}_{t-2}} - \nabla_{\boldsymbol{\theta}} L(\widetilde{\boldsymbol{\theta}}_{t-1})$, for $t = 1, 2, \ldots$,

where $\widetilde{\eta}_t = P_t P_{t+1} \eta_t$, $P_t = \prod_{i=-1}^{t} \alpha_i^{-1}$, $\forall t \geq -1$ and $\alpha_t$ recursively defined as

$$\alpha_t = -\eta_{t-1} \lambda_{t-1} + 1 + \frac{\eta_{t-1}}{\eta_{t-2}} \gamma (1 - \alpha_{t-1}^{-1}), \forall t \geq 1. \tag{7}$$

The LR schedule $\{\widetilde{\eta}_t\}_{t=0}^{\infty}$ is called *Tapered Exponential ++*, or *TEXP++*.

## 3 EXAMPLE ILLUSTRATING INTERPLAY OF WD AND BN

The paper so far has shown that effects of different hyperparameters in training are not easily separated, since their combined effect on the trajectory is complicated. We give a simple example to illustrate this, where convergence is guaranteed if we use either BatchNorm or weight decay in isolation, but convergence fails if both are used. (Momentum is turned off for clarity of presentation)

**Setting:** Suppose we are fine-tuning the last linear layer of the network, where the input of the last layer is assumed to follow a standard Gaussian distribution $\mathcal{N}(0, I_m)$, and $m$ is the input dimension of last layer. We also assume this is a binary classification task with logistic loss, $l(u, y) = \ln(1 + \exp(-uy))$, where label $y \in \{-1, 1\}$ and $u \in \mathbb{R}$ is the output of the neural network. The training algorithm is SGD with constant LR and WD, and without momentum. For simplicity we assume the batch size $B$ is very large so we could assume the covariance of each batch $\mathcal{B}_t$ concentrates and is approximately equal to identity, namely $\frac{1}{B} \sum_{i=1}^{B} \boldsymbol{x}_{t,b} \boldsymbol{x}_{t,b}^{\top} \approx I_m$. We also assume the the input of the last layer are already separable, and w.l.o.g. we assume the label is equal to the sign of the first coordinate of $\boldsymbol{x} \in \mathbb{R}^m$, namely $\text{sign}(x_1)$. Thus the training loss and training error are simply

$$L(\boldsymbol{w}) = \mathbb{E}_{\boldsymbol{x} \sim \mathcal{N}(0, I_m), y = \text{sign}(x_1)} \left[ \ln(1 + \exp(-\boldsymbol{x}^{\top} \boldsymbol{w} y)) \right], \quad \Pr_{\boldsymbol{x} \sim \mathcal{N}(0, I_m), y = \text{sign}(x_1)} \left[ \boldsymbol{x}^{\top} \boldsymbol{w} y \leq 0 \right] = \frac{1}{\pi} \arccos \frac{w_1}{\|\boldsymbol{w}\|}$$

*Case 1: WD alone:* Since both the above objective with L2 regularization is strongly convex and smooth in $\boldsymbol{w}$, vanilla GD *with suitably small learning rate* could get arbitrarily close to the global minimum for this regularized objective. In our case, large batch SGD behaves similarly to GD and can achieve $O(\sqrt{\frac{\eta\lambda}{B}})$ test error following the standard analysis of convex optimization.

*Case 2: BN alone:* Add a BN layer after the linear layer, and fix scalar and bias term to 1 and 0. The objective becomes

$$L_{BN}(\boldsymbol{w}) = \mathbb{E}_{\boldsymbol{x} \sim \mathcal{N}(0, I_m), y = \text{sign}(x_1)} [L_{BN}(\boldsymbol{w}, \boldsymbol{x})] = \mathbb{E}_{\boldsymbol{x} \sim \mathcal{N}(0, I_m), y = \text{sign}(x_1)} \left[ \ln(1 + \exp(-\boldsymbol{x}^{\top} \frac{\boldsymbol{w}}{\|\boldsymbol{w}\|} y)) \right].$$

From Appendix B.6, there's some constant $C$, such that $\forall \boldsymbol{w} \in \mathbb{R}^m$ with constant probability, $\|\nabla_{\boldsymbol{w}} L_{BN}(\boldsymbol{w}, \boldsymbol{x})\| \geq \frac{C}{\|\boldsymbol{w}\|}$. By Pythagorean Theorem, $\|\boldsymbol{w}_{t+1}\|^4 = (\|\boldsymbol{w}_t\|^2 + \eta^2 \|\nabla_{\boldsymbol{w}} L_{BN}(\boldsymbol{w}_t, \boldsymbol{x})\|^2)^2 \geq \|\boldsymbol{w}_t\|^4 + 2\eta^2 \|\boldsymbol{w}_t\|^2 \|\nabla_{\boldsymbol{w}} L_{BN}(\boldsymbol{w}_t, \boldsymbol{x})\|^2$. As a result, for any *fixed learning rate*, $\|\boldsymbol{w}_{t+1}\|^4 \geq 2 \sum_{i=1}^{t} \eta^2 \|\boldsymbol{w}\|^2 \|\nabla_{\boldsymbol{w}} L_{BN}(\boldsymbol{w}_i, \boldsymbol{x})\|^2$ grows at least linearly with high probability. Following the analysis of Arora et al. (2019), this is like reducing the effective learning

rate, and when $\|\boldsymbol{w}_t\|$ is large enough, the effective learning rate is small enough, and thus SGD can find the local minimum, which is the unique global minimum.

*Case 3: Both BN and WD:* Suppose WD factor is $\lambda$, LR is $\eta$, we have the following theorem:

**Theorem 3.1.** [Nonconvergence] Starting from iteration any $T_0$, with probability $1 - \delta$ over the randomness of samples, the training error will be larger than $\frac{\varepsilon}{\pi}$ at least once for the following consecutive $\frac{1}{2(\eta\lambda - 2\varepsilon^2)} \ln \frac{64\|w_{T_0}\|^2 \varepsilon \sqrt{B}}{\eta \sqrt{m-2}} + 9 \ln \frac{1}{\delta}$ iterations.

*Sketch.* (See full proof in Appendix B.) The high level idea of this proof is that if the test error is low, the weight is restricted in a small cone around the global minimum, and thus the amount of the gradient update is bounded by the size of the cone. In this case, the growth of the norm of the weight by Pythagorean Theorem is not large enough to cancel the shrinkage brought by weight decay. As a result, the norm of the weight converges to 0 geometrically. Again we need to use the lower bound for size of the gradient, that $\|\nabla_{\boldsymbol{w}} L_t\| = \Theta(\frac{\eta\sqrt{m}}{\|\boldsymbol{w}_t\|})$ holds with constant probability. Thus the size of the gradient will grow along with the shrinkage of $\|\boldsymbol{w}_t\|$ until they're comparable, forcing the weight to leave the cone in next iteration. □

## 4 EXPERIMENTS

The translation to exponential LR schedule is exact except for one-time momentum correction term entering new phases. The experiments explore the effect of this correction term. The Tapered Exponential(TEXP) LR schedule contains two parts when entering a new phase I: an instant LR decay ($\frac{\eta_I}{\eta_{I-1}}$) and an adjustment of the growth factor ($\alpha^*_{I-1} \to \alpha^*_I$). The first part is relative small compared to the huge exponential growing. Thus a natural question arises: *Can we simplify TEXP LR schedule by dropping the part of instant LR decay?*

Also, previously we have only verified our equivalence theorem in Step Decay LR schedules. But it's not sure how would the Exponential LR schedule behave on more rapid time-varying LR schedules such as Cosine LR schedule.

**Settings:** We train PreResNet32 on CIFAR10. The initial learning rate is 0.1 and the momentum is 0.9 in all settings. We fix all the scalar and bias of BN, because otherwise they together with the following conv layer grow exponentially, sometimes exceeding the range of Float32 when trained with large growth rate for a long time. We fix the parameters in the last fully connected layer for scale invariance of the objective.

### 4.1 THE BENEFIT OF INSTANT LR DECAY

We tried the following LR schedule (we call it *TEXP--*). Interestingly, up to correction of momentum when entering a new phase, this schedule is equivalent to a constant LR schedule, but with the weight decay coefficient reduced correspondingly at the start of each phase. (See Theorem B.2 and Figure 6)

$$\text{TEXP--:} \quad \widetilde{\eta}_{t+1} = \begin{cases} \widetilde{\eta}_t \times (\alpha^*_{I-1})^{-2} & \text{if } T_{I-1} + 1 \leq t \leq T_I - 1, I \geq 1; \\ \widetilde{\eta}_t \times (\alpha^*_I)^{-1}(\alpha^*_{I-1})^{-1} & \text{if } t = T_I, I \geq 1, \end{cases} \tag{8}$$

where $\alpha^*_I = \frac{1 + \gamma - \lambda\eta^*_I + \sqrt{(1 + \gamma - \lambda\eta^*_I)^2 - 4\gamma}}{2}$, $\widetilde{\eta}_0 = \eta_0 \cdot (\alpha^*_0)^{-1} = \eta^*_0 \cdot (\alpha^*_0)^{-1}$.

### 4.2 BETTER EXPONENTIAL LR SCHEDULE WITH COSINE LR

We applied the TEXP LR schedule (Theorem 2.12) on the Cosine LR schedule (Loshchilov & Hutter, 2016), where the learning rate changes every epoch, and thus correction terms cannot be ignored. The LR at epoch $t \leq T$ is defined as: $\eta_t = \eta_0 \frac{1 + \cos(\frac{t}{T}\pi)}{2}$. Our experiments show this hybrid schedule with Cosine LR performs better on CIFAR10 than Step Decay, but this finding needs to be verified on other datasets.

## 5 CONCLUSIONS

The paper shows rigorously how BN allows a host of very exotic learning rate schedules in deep learning, and verifies these effects in experiments. The lr increases exponentially in almost every

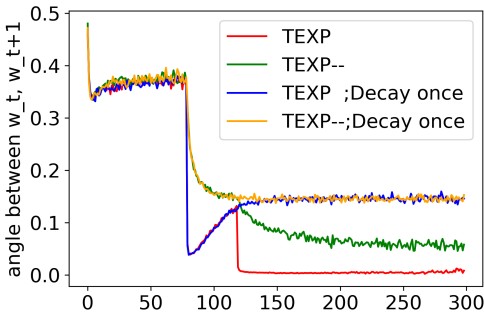 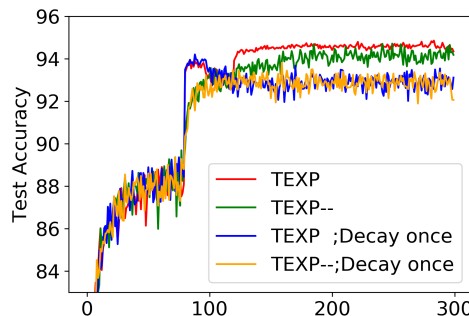

**Figure 3:** Instant LR decay is crucial when LR growth $\widetilde{\eta}_t/\widetilde{\eta}_{t-1}-1$ is very small. The original LR of Step Decay is decayed by 10 at epoch 80, 120 respectively. In the third phase, LR growth $\widetilde{\eta}_t/\widetilde{\eta}_{t-1}-1$ is approximately 100 times smaller than that in the third phase, it would take TEXP-- hundreds of epochs to reach its equilibrium. As a result, TEXP achieves better test accuracy than TEXP--. As a comparison, in the second phase, $\widetilde{\eta}_t/\widetilde{\eta}_{t-1}-1$ is only 10 times smaller than that in the first phase and it only takes 70 epochs to return to equilibrium.

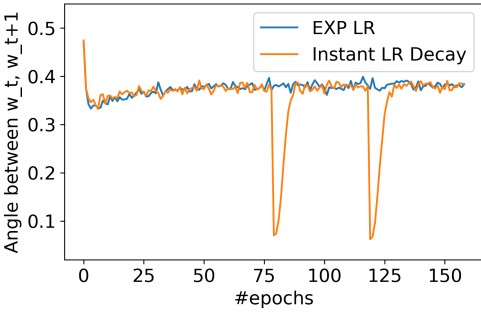 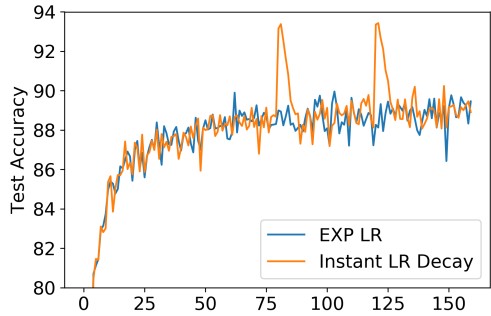

**Figure 4:** Instant LR decay has only temporary effect when LR growth $\widetilde{\eta}_t/\widetilde{\eta}_{t-1}-1$ is large. The blue line uses an exponential LR schedule with constant exponent. The orange line multiplies its LR by the same constant each iteration, but also divide LR by 10 at the start of epoch 80 and 120. The instant LR decay only allows the parameter to stay at good local minimum for 1 epoch and then diverges, behaving similarly to the trajectories without no instant LR decay.

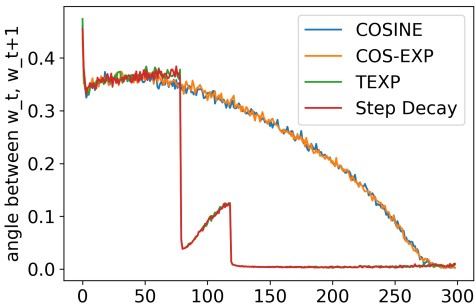 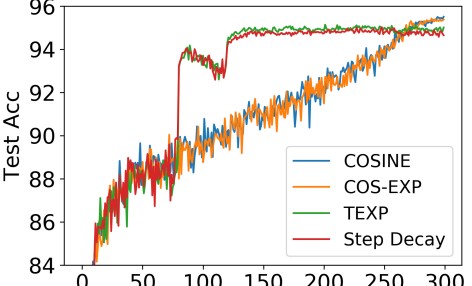

**Figure 5:** Both Cosine and Step Decay schedule behaves almost the same as their exponential counterpart, as predicted by our equivalence theorem. The (exponential) Cosine LR schedule achieves better test accuracy, with a entirely different trajectory.

iteration during training. The exponential increase derives from use of weight decay, but the precise expression involves momentum as well. We suggest that the efficacy of this rule may be hard to explain with canonical frameworks in optimization.

Our analyses of BN is a substantial improvement over earlier theoretical analyses, since it accounts for weight decay and momentum, which are always combined in practice.

Our tantalising experiments with a hybrid of exponential and cosine rates suggest that more surprises may lie out there. Our theoretical analysis of interrelatedness of hyperparameters could also lead to faster hyperparameter search.

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

## A  VIEWING EXP LR VIA CANONICAL OPTIMIZATION FRAMEWORK

This section tries to explain why the efficacy of exponential LR in deep learning is mysterious to us, at least as viewed in the canonical framework of optimization theory.

*Canonical framework for analysing 1st order methods* This focuses on proving that each —or most—steps of GD noticeably reduce the objective, by relying on some assumption about the spectrum norm of the hessian of the loss, and most frequently, the *smoothness*, denoted by $\beta$. Specifically, for GD update $\boldsymbol{\theta}_{t+1} = \boldsymbol{\theta}_t - \eta\nabla L(\boldsymbol{\theta}_t)$, we have

$$L(\boldsymbol{\theta}_{t+1}) - L(\boldsymbol{\theta}_t) \leq (\boldsymbol{\theta}_{t+1} - \boldsymbol{\theta}_t)^\top \nabla L(\boldsymbol{\theta}_t) + \frac{\beta}{2}\|\boldsymbol{\theta}_{t+1} - \boldsymbol{\theta}_t\|^2 = -\eta(1 - \frac{\beta\eta}{2})\|\nabla L(\boldsymbol{\theta}_t)\|^2.$$

When $\beta < \frac{2}{\eta}$, the first order term is larger than the second order one, guaranteeing the loss value decreases. Since the analysis framework treats the loss as a black box (apart from the assumed bounds on the derivative norms), and the loss is non-convex, the best one can hope for is to prove speedy convergence to a stationary point (where gradient is close to 0). An increasing body of work proves such results.

Now we turn to difficulties in understanding the exponential LR in context of the above framework and with scale-invariance in the network.

1. Since loss is same for $\boldsymbol{\theta}$ and $c \cdot \boldsymbol{\theta}$ for all $c > 0$ a simple calculation shows that along any straight line through the origin, smoothness is a decreasing function of $c$, and is very high close to origin. (Note: it is also possible to one can show the following related fact: In any ball containing the origin, the loss is nonconvex.)

   Thus if one were trying to apply the canonical framework to argue convergence to a stationary point, the natural idea would be to try to grow the norm of the parameters until smoothness drops enough that the above-mentioned *Canonical Framework* starts to apply. Arora et al. (2019) showed this happens in GD with fixed LR (WD turned off), and furthermore the resulting convergence rate to stationary point is asymptotically similar to analyses of nonconvex optimization with learning rate set as in the *Canonical framework*. Santurkar et al. (2018) observed similar phenomenon in experiments, which they described as a smoothening of the objective due to BN.

2. The *Canonical Framework* can be thought of as a discretization of continuous gradient descent (i.e., gradient flow): in principle it is possible to use arbitrarily small learning rate, but one uses finite learning rate merely to keep the number of iterations small. The discrete process approximates the continuous process due to smoothness being small.

   In case of gradient flow with weight decay (equivalently, with exponential LR schedule) the discrete process cannot track the continuous process for very long, which suggests that any explanation of the benefits of exponential LR may need to rely on discrete process being somehow better. The reason being that for gradient flow one can decouple the speed of the $\boldsymbol{\theta}_t$ into the tangential and the radial components, where the former one has no effect on the norm and the latter one has no effect on the objective but scales the tangential gradient exponentially. Thus the Gradient Flow with WD gives exactly the same trajectory as vanilla Gradient Flow does, excepting a exponential reparametrization with respect to time $t$.

3. It can be shown that if the local smoothness is upperbounded by $\frac{2}{\eta}$ (as stipulated in *Canonical Framework*) during a sequence $\boldsymbol{\theta}_t$ ($t = 1, 2, \ldots$) of GD updates with WD and constant LR then such sequence satisfies $\boldsymbol{\theta}_t \to \mathbf{0}$. This contrasts with the usual experimental observation that $\boldsymbol{\theta}_t$ stays bounded away from $\mathbf{0}$. One should thus conclude that in practice, with constant LR and WD, smoothness doesn't always stay small (unlike the above analyses where WD is turned off).

## B  OMITTED PROOFS

### B.1  OMITTED PROOF IN SECTION 2

**Lemma B.1** (Some Facts about Equation 1). *Suppose $z^1, z^2(z^1 \geq z^2)$ are the two real roots of the the following equation, we have*

$$x^2 - (1 + \gamma - \lambda\eta)x + \gamma = 0$$

1. $z^1 = \frac{1+\gamma-\lambda\eta+\sqrt{(1-\gamma)^2-2(1+\gamma)\lambda\eta+\lambda^2\eta^2}}{2}$, $z^2 = \frac{1+\gamma-\lambda\eta-\sqrt{(1-\gamma)^2-2(1+\gamma)\lambda\eta+\lambda^2\eta^2}}{2}$

2. $z^1, z^2$ *are real* $\iff \lambda\eta \le (1-\sqrt{\gamma})^2$;

3. $z^1 z^2 = \gamma, z^1 + z^2 = (1+\gamma-\lambda\eta)$;

4. $\gamma \le z^2 \le z^1 \le 1$;

5. *Let* $t = \frac{\lambda\eta}{1-\gamma}$, *we have* $z^1 \ge \frac{1}{1+t} \ge 1 - t = 1 - \frac{\lambda\eta}{1-\gamma}$.

6. *if we view* $z^1(\lambda\eta), z^2(\lambda\eta)$ *as functions of* $\lambda\eta$, *then* $z^1(\lambda\eta)$ *is monotone decreasing,* $z^2(\eta)$ *is monotone increasing.*

*Proof.*

4. Let $f(x) = x^2 - (1+\gamma-\lambda\eta)x + \gamma$, we have $f(1) = f(\gamma) = \lambda\eta \ge 0$. Note the minimum of $f$ is taken at $x = \frac{1+\gamma-\lambda\eta}{2} \in [0,1]$, the both roots of $f(x) = 0$ must lie between 0 and 1, if exists.

5

$$
\begin{aligned}
1 - z^1 &= \frac{1-\gamma+\lambda\eta-\sqrt{(1-\gamma)^2-2(1+\gamma)\lambda\eta+\lambda^2\eta^2}}{2} \\
&= (1-\gamma)\frac{1+t-\sqrt{1-\frac{1+\gamma}{1-\gamma}t+t^2}}{2} \\
&= (1-\gamma)\frac{2t+2\frac{1+\gamma}{1-\gamma}t}{2(1+t+\sqrt{1-\frac{1+\gamma}{1-\gamma}t+t^2})} \\
&\le (1-\gamma)\frac{\frac{4}{1-\gamma}t}{4(1+t)} \\
&= \frac{t}{(1+t)}
\end{aligned}
$$

6. Note that $(z^1 - z^2)^2 = (z^1 + z^2)^2 - 4z^1 z^2 = (1+\gamma-\lambda\eta)^2 - 4\gamma$ is monotone decreasing, since $z^1(\lambda\eta) + z^2(\lambda\eta)$ is constant, $z^1(\lambda\eta) \ge z^2(\lambda\eta)$, $z^1(\lambda\eta)$ must be decreasing and $z^2(\lambda\eta)$ must be increasing.

$\square$

## B.2 OMITTED PROOFS IN SECTION 2.1

*Proof of Lemma 2.2.* For any $(\boldsymbol{\theta}, \eta)$, we have

$$
\text{GD}_t^\rho(\boldsymbol{\theta}, \eta) = (\rho\boldsymbol{\theta} - \eta\nabla L_t(\boldsymbol{\theta}), \eta) = [\Pi_1^\rho \circ \Pi_2^\rho \circ \text{GD}_t](\boldsymbol{\theta}, \frac{\eta}{\rho}) = [\Pi_1^\rho \circ \Pi_2^\rho \circ \text{GD}_t \circ \Pi_2^{\rho^{-1}}](\boldsymbol{\theta}, \eta).
$$

*Proof of Lemma 2.4.* For any $(\boldsymbol{\theta}, \eta)$, we have

$$
\left[\text{GD}_t \circ \Pi_1^c \circ \Pi_2^{c^2}\right](\boldsymbol{\theta}, \eta) = \text{GD}_t(c\boldsymbol{\theta}, c^2\eta) = (c\boldsymbol{\theta} - c^2\boldsymbol{\theta}\nabla L_t(c\boldsymbol{\theta}), c^2\eta) \overset{*}{=} (c(\boldsymbol{\theta} - \nabla L_t(\boldsymbol{\theta})), c^2\eta)
$$
$$
= \left[\Pi_1^c \circ \Pi_2^{c^2} \circ \text{GD}_t\right](\boldsymbol{\theta}, \eta). \qquad\qquad (\overset{*}{=}: \text{Scale Invariance, Lemma } 1.3) \qquad\qquad \square
$$

## B.3 OMITTED PROOFS IN SECTION 2.2

*Proof of Lemma 2.7.* For any input $(\boldsymbol{\theta}, \eta, \boldsymbol{\theta}', \eta')$, it's easy to check both composed maps have the same outputs on the 2,3,4th coordinates, namely $(c^2\eta, c\boldsymbol{\theta}, c^2\eta')$. For the first coordinate, we have

$$\left[\text{GD}^\rho(c\boldsymbol{\theta}, c^2\eta, c\boldsymbol{\theta}', c^2\eta)\right]_1 = \rho c\boldsymbol{\theta} + c^2\eta\left(\gamma\frac{\boldsymbol{\theta}-\boldsymbol{\theta}'}{\eta'} - \nabla L_t(c\boldsymbol{\theta})\right) \overset{*}{=} c\left(\boldsymbol{\theta} + \eta\left(\gamma\frac{\boldsymbol{\theta}-\boldsymbol{\theta}'}{\eta'} - \nabla L_t(\boldsymbol{\theta})\right)\right)$$

$$=c\left[\text{GD}^\rho(\boldsymbol{\theta}, \eta, \boldsymbol{\theta}', \eta)\right]_1. \qquad \overset{*}{=}: \text{Scale Invariance, Lemma 1.3} \qquad \square$$

*Proof of Lemma 2.8.* For any input $(\boldsymbol{\theta}, \eta, \boldsymbol{\theta}', \eta')$, it's easy to check both composed maps have the same outputs on the 2,3,4th coordinates, namely $(\eta, \boldsymbol{\theta}, \eta)$. For the first coordinate, we have

$$\left[\left[\Pi_3^\alpha \circ \Pi_4^\alpha \circ \Pi_2^\alpha \circ \text{GD}_t \circ \Pi_2^{\alpha^{-1}} \circ \Pi_3^\alpha \circ \Pi_4^\alpha\right](\boldsymbol{\theta}, \eta, \boldsymbol{\theta}', \eta)\right]_1 = \alpha\left[\text{GD}_t(\boldsymbol{\theta}, \alpha^{-1}\eta, \alpha\boldsymbol{\theta}', \alpha\eta)\right]_1$$

$$=\alpha\left(\boldsymbol{\theta} + \alpha^{-1}\eta\left(\gamma\frac{\boldsymbol{\theta}-\boldsymbol{\theta}'}{\eta} - \nabla L_t(\boldsymbol{\theta})\right)\right)$$

$$= \left(\alpha + \gamma\alpha^{-1}\right)\boldsymbol{\theta} - \eta\nabla L_t(\boldsymbol{\theta}) - \eta\gamma\frac{\boldsymbol{\theta}'}{\eta}$$

$$= (\rho + \gamma)\boldsymbol{\theta} - \eta\nabla L_t(\boldsymbol{\theta}) - \gamma\boldsymbol{\theta}' \quad = \left[\text{GD}_t^\rho(\boldsymbol{\theta}, \eta, \boldsymbol{\theta}', \eta)\right]_1 \qquad \square$$

## B.4 OMITTED PROOFS OF THEOREM 2.12

In this subsection we will prove a stronger version of Theorem 2.12(restated below), allowing the WD,$\lambda_I$ changing each phase.

**Theorem B.2** (A stronger version of Theorem 2.12). There exists a way to correct the momentum only at the first iteration of each phase, such that the following Tapered-Exponential LR schedule (TEXP) $\{\widetilde{\eta}_t\}$ with momentum factor $\gamma$ and no WD, leads the same sequence networks in function space compared to that of Step Decay LR schedule(Definition 2.11) with momentum factor $\gamma$ and *phase-dependent* WD $\lambda_I^*$ in phase $I$, where phase $I$ lasts from iteration $T_I$ to iteration $T_{I+1}, T_0 = 0$.

$$\widetilde{\eta}_{t+1} = \begin{cases} \widetilde{\eta}_t \times (\alpha_{I-1}^*)^{-2} & \text{if } T_{I-1}+1 \leq t \leq T_I - 1, I \geq 1, \\ \widetilde{\eta}_t \times \frac{\eta_I^*}{\eta_{I-1}^*} \times (\alpha_I^*)^{-1}(\alpha_{I-1}^*)^{-1} & \text{if } t = T_I, I \geq 1 \end{cases}, \qquad (9)$$

where $\alpha_I^* = \frac{1+\gamma-\lambda_I^*\eta_I^*+\sqrt{\left(1+\gamma-\lambda_I^*\eta_I^*\right)^2-4\gamma}}{2}, \widetilde{\eta}_0 = \eta_0(\alpha_0^*)^{-1} = \eta_0^*(\alpha_0^*)^{-1}$.

Towards proving Theorem 2.12, we need the following lemma which holds by expanding the definition, and we omit its proof.

**Lemma B.3** (Canonicalization). *We define the* Canonicalization *map as* $N(\boldsymbol{\theta}, \eta, \boldsymbol{\theta}', \eta') = (\boldsymbol{\theta}, \eta, \boldsymbol{\theta} - \frac{\eta}{\eta'}(\boldsymbol{\theta} - \boldsymbol{\theta}'), \eta)$, *and it holds that*

1. $GD_t^\rho \circ N = GD_t^\rho, \forall \rho > 0, t \geq 0$.

2. $N \circ \left[\Pi_1^c \circ \Pi_2^{c^2} \circ \Pi_3^c \circ \Pi_4^{c^2}\right] = \left[\Pi_1^c \circ \Pi_2^{c^2} \circ \Pi_3^c \circ \Pi_4^{c^2}\right] \circ N, \forall c > 0$.

Similar to the case of momentum-free SGD, we define the notion of equivalent map below

**Definition B.4** (Equivalent Maps). For two maps $F$ and $G$, we say $F$ is equivalent to $G$ iff $\exists c > 0$, $F = \left[\Pi_1^c \circ \Pi_2^{c^2} \circ \Pi_3^c \circ \Pi_4^{c^2}\right] \circ G$, which is also denoted by $F \overset{c}{\sim} G$.

Note that for any $(\boldsymbol{\theta}, \eta, \boldsymbol{\theta}', \eta')$, $[N(\boldsymbol{\theta}, \eta, \boldsymbol{\theta}', \eta')]_2 = [N(\boldsymbol{\theta}, \eta, \boldsymbol{\theta}', \eta')]_4$. Thus as a direct consequence of Lemma 2.8, the following lemma holds.

**Lemma B.5.** $\forall \rho, \alpha > 0$, $GD_t^\rho \circ N \overset{\alpha}{\sim} \Pi_3^{\alpha^{-1}} \circ \Pi_4^{\alpha^{-1}} \circ \Pi_2^{\alpha^{-1}} \circ GD_t \circ \Pi_2^{\alpha^{-1}} \circ \Pi_3^\alpha \circ \Pi_4^\alpha \circ N$.

*Proof of Theorem 2.12.* Starting with initial state $(\boldsymbol{\theta}_0, \eta_0, \boldsymbol{\theta}_{-1}, \eta_{-1})$ where $\eta_{-1} = \eta_0$ and a given LR schedule $\{\eta_t\}_{t\geq 0}$, the parameters generated by GD with WD and momentum satisfies the following relationship:

$$(\boldsymbol{\theta}_{t+1}, \eta_{t+1}, \boldsymbol{\theta}_t, \eta_t) = \left[ \Pi_2^{\frac{\eta_{t+1}}{\eta_t}} \circ \mathrm{GD}_t^{1 - \eta_t \lambda_t} \right] (\boldsymbol{\theta}_t, \eta_t, \boldsymbol{\theta}_{t-1}, \eta_{t-1}).$$

Define $\bigcirc_{t=a}^{b} F_t = F_b \circ F_{b-1} \circ \ldots \circ F_a$, for $a \le b$. By Lemma B.3 and Lemma B.5, letting $\alpha_t$ be the root of $x^2 - (\gamma + 1 - \eta_{t-1} \lambda_{t-1}) x + \gamma = 0$, we have

$$\begin{aligned}
&\bigcirc_{t=0}^{T-1} \left[ \Pi_2^{\frac{\eta_{t+1}}{\eta_t}} \circ \mathrm{GD}_t^{1 - \eta_t \lambda_t} \right] \\
=&\bigcirc_{t=0}^{T-1} \left[ \Pi_2^{\frac{\eta_{t+1}}{\eta_t}} \circ \mathrm{GD}_t^{1 - \eta_t \lambda_t} \circ N \right] \\
\overset{\prod_{i=0}^{T-1} \alpha_i}{\sim}&\bigcirc_{t=0}^{T-1} \left[ \Pi_2^{\frac{\eta_{t+1}}{\eta_t}} \circ \Pi_3^{\alpha_{t+1}^{-1}} \circ \Pi_4^{\alpha_{t+1}^{-1}} \circ \Pi_2^{\alpha_{t+1}^{-1}} \circ \mathrm{GD}_t \circ \Pi_2^{\alpha_{t+1}^{-1}} \circ \Pi_3^{\alpha_{t+1}} \circ \Pi_4^{\alpha_{t+1}} \circ N \right] \\
=&\Pi_2^{\frac{\eta_T}{\eta_{T-1}}} \circ \Pi_3^{\alpha_T^{-1}} \circ \Pi_4^{\alpha_T^{-1}} \circ \Pi_2^{\alpha_T^{-1}} \circ \mathrm{GD}_{T-1} \circ \left( \bigcirc_{t=1}^{T-1} \left[ \Pi_2^{\alpha_{t+1}^{-1} \alpha_t^{-1}} \circ H_t \circ \mathrm{GD}_{t-1} \right] \right) \circ \Pi_2^{\alpha_1^{-1}} \circ \Pi_3^{\alpha_1} \circ \Pi_4^{\alpha_1} \circ N,
\end{aligned}$$
(10)

where $\overset{\prod_{i=0}^{T-1} \alpha_i}{\sim}$ is because of Lemma B.5, and $H_t$ is defined as

$$H_t = \Pi_2^{\alpha_t} \circ \Pi_2^{\frac{\eta_{t-1}}{\eta_t}} \circ \Pi_3^{\alpha_{t+1}} \circ \Pi_4^{\alpha_{t+1}} \circ N \circ \Pi_3^{\alpha_t^{-1}} \circ \Pi_4^{\alpha_t^{-1}} \circ \Pi_2^{\alpha_t^{-1}} \circ \Pi_2^{\frac{\eta_t}{\eta_{t-1}}}.$$

Since the canonicalization map $N$ only changes the momentum part of the state, it's easy to check that $H_t$ doesn't touch the current parameter $\boldsymbol{\theta}$ and the current LR $\eta$. Thus $H_t$ only changes the momentum part of the input state. Now we claim that $H_t \circ \mathrm{GD}_{t-1} = \mathrm{GD}_{t-1}$ whenever $\eta_t = \eta_{t-1}$. This is because when $\eta_t = \eta_{t-1}$, $\alpha_t = \alpha_{t+1}$, thus $H_t \circ \mathrm{GD}_{t-1} = \mathrm{GD}_{t-1}$. In detail,

$$\begin{aligned}
&H_t \circ \mathrm{GD}_{t-1} \\
=&\Pi_2^{\alpha_t} \circ \Pi_3^{\alpha_t} \circ \Pi_4^{\alpha_t} \circ N \circ \Pi_3^{\alpha_t^{-1}} \circ \Pi_4^{\alpha_t^{-1}} \circ \Pi_2^{\alpha_t^{-1}} \circ \mathrm{GD}_{t-1} \\
\overset{*}{=}&\Pi_2^{\alpha_t} \circ \Pi_3^{\alpha_t} \circ \Pi_4^{\alpha_t} \circ \Pi_3^{\alpha_t^{-1}} \circ \Pi_4^{\alpha_t^{-1}} \circ \Pi_2^{\alpha_t^{-1}} \circ \mathrm{GD}_{t-1} \\
=&\mathrm{GD}_{t-1},
\end{aligned}$$

where $\overset{*}{=}$ is because GD update $\mathrm{GD}_t$ sets $\eta'$ the same as $\eta$, and thus ensures the input of $N$ has the same momentum factor in buffer as its current momentum factor, which makes $N$ an identity map.

Thus we could rewrite Equation 10 with a "sloppy" version of $H_t$, $H_t' = \begin{cases} H_t & \eta_t \ne \eta_{t-1}; \\ Id & \text{o.w.} \end{cases}$:

$$\begin{aligned}
&\bigcirc_{t=0}^{T-1} \left[ \Pi_2^{\frac{\eta_{t+1}}{\eta_t}} \circ \mathrm{GD}_t^{1 - \eta_t \lambda_t} \right] \\
=&\Pi_2^{\frac{\eta_T}{\eta_{T-1}}} \circ \Pi_3^{\alpha_T^{-1}} \circ \Pi_4^{\alpha_T^{-1}} \circ \Pi_2^{\alpha_T^{-1}} \circ \mathrm{GD}_{T-1} \circ \left( \bigcirc_{t=1}^{T-1} \left[ \Pi_2^{\alpha_{t+1}^{-1} \alpha_t^{-1}} \circ H_t' \circ \mathrm{GD}_{t-1} \right] \right) \circ \Pi_2^{\alpha_1^{-1}} \circ \Pi_3^{\alpha_1} \circ \Pi_4^{\alpha_1} \circ N \\
=&\Pi_2^{\frac{\eta_T}{\eta_{T-1}}} \circ \Pi_3^{\alpha_T^{-1}} \circ \Pi_4^{\alpha_T^{-1}} \circ \Pi_2^{\alpha_T^{-1}} \circ \left( \bigcirc_{t=1}^{T-1} \left[ \mathrm{GD}_t \circ \Pi_2^{\alpha_{t+1}^{-1} \alpha_t^{-1}} \circ H_t' \right] \right) \circ \mathrm{GD}_0 \circ \Pi_2^{\alpha_1^{-1}} \circ \Pi_3^{\alpha_1} \circ \Pi_4^{\alpha_1} \circ N,
\end{aligned}$$
(11)

Now we construct the desired sequence of parameters achieved by using the Tapered Exp LR schedule 9 and the additional one-time momentum correction per phase. Let $(\widetilde{\boldsymbol{\theta}}_0, \widetilde{\eta}_0, \widetilde{\boldsymbol{\theta}}_{-1}, \widetilde{\eta}_{-1}) = (\boldsymbol{\theta}_0, \eta_0, \boldsymbol{\theta}_{-1}, \eta_0)$, and

$$(\widetilde{\boldsymbol{\theta}}_1, \widetilde{\eta}_1, \widetilde{\boldsymbol{\theta}}_0, \widetilde{\eta}_0) = \left[ \text{GD}_0 \circ \Pi_2^{\alpha_1^{-1}} \circ \Pi_3^{\alpha_1} \circ \Pi_4^{\alpha_1} \circ N \right] (\widetilde{\boldsymbol{\theta}}_0, \widetilde{\eta}_0, \widetilde{\boldsymbol{\theta}}_{-1}, \widetilde{\eta}_{-1})$$

$$= \left[ \text{GD}_0 \circ \Pi_2^{\alpha_1^{-1}} \circ \Pi_3^{\alpha_1} \circ \Pi_4^{\alpha_1} \right] (\widetilde{\boldsymbol{\theta}}_0, \widetilde{\eta}_0, \widetilde{\boldsymbol{\theta}}_{-1}, \widetilde{\eta}_{-1});$$

$$(\widetilde{\boldsymbol{\theta}}_{t+1}, \widetilde{\eta}_{t+1}, \widetilde{\boldsymbol{\theta}}_t, \widetilde{\eta}_t) = \left[ \text{GD}_t \circ \Pi_2^{\alpha_{t+1}^{-1}\alpha_t^{-1}} \circ H_t' \right] (\widetilde{\boldsymbol{\theta}}_t, \widetilde{\eta}_t, \widetilde{\boldsymbol{\theta}}_{t-1}, \widetilde{\eta}_{t-1}).$$

we claim $\{\widetilde{\boldsymbol{\theta}}_t\}_{t=0}$ is the desired sequence of parameters. We've already shown that $\boldsymbol{\theta}_t \sim \widetilde{\boldsymbol{\theta}}_t$, $\forall t$. Clearly $\{\widetilde{\boldsymbol{\theta}}_t\}_{t=0}$ is generated using only vanilla GD, scaling LR and modifying the momentum part of the state. When $t \neq T_I$ for any $I$, $\eta_t = \eta_{t-1}$ and thus $H_t' = Id$. Thus the modification on the momentum could only happen at $T_I (I \geq 0)$. Also it's easy to check that $\alpha_t = \alpha_I^*$, if $T_I + 1 \leq t \leq T_{I+1}$. □

### B.5 OMITTED PROOFS OF THEOREM 2.13

**Theorem B.6.** The following two sequences of parameters ,$\{\boldsymbol{\theta}_t\}_{t=0}^{\infty}$ and $\{\widetilde{\boldsymbol{\theta}}_t\}_{t=0}^{\infty}$, define the same sequence of network functions, i.e. $f_{\boldsymbol{\theta}_t} = f_{\widetilde{\boldsymbol{\theta}}_t}$, $\forall t \in \mathbb{N}$, given the initial conditions, $\widetilde{\boldsymbol{\theta}}_0 = P_0\boldsymbol{\theta}_0$, $\widetilde{\boldsymbol{\theta}}_{-1} = P_{-1}\boldsymbol{\theta}_{-1}$.

1. $\frac{\boldsymbol{\theta}_t - \boldsymbol{\theta}_{t-1}}{\eta_{t-1}} = \gamma\frac{\boldsymbol{\theta}_{t-1} - \boldsymbol{\theta}_{t-2}}{\eta_{t-2}} - \nabla_{\boldsymbol{\theta}}\left( (L(\boldsymbol{\theta}_{t-1}) + \frac{\lambda_{t-1}}{2}\|\boldsymbol{\theta}_{t-1}\|_2^2) \right)$, for $t = 1, 2, \ldots$;

2. $\frac{\widetilde{\boldsymbol{\theta}}_t - \widetilde{\boldsymbol{\theta}}_{t-1}}{\widetilde{\eta}_{t-1}} = \gamma\frac{\widetilde{\boldsymbol{\theta}}_{t-1} - \widetilde{\boldsymbol{\theta}}_{t-2}}{\widetilde{\eta}_{t-2}} - \nabla_{\boldsymbol{\theta}}L(\widetilde{\boldsymbol{\theta}}_{t-1})$, for $t = 1, 2, \ldots$,

where $\widetilde{\eta}_t = P_t P_{t+1}\eta_t$, $P_t = \prod\limits_{i=-1}^{t} \alpha_i^{-1}, \forall t \geq -1$ and $\alpha_t$ recursively defined as

$$\alpha_t = -\eta_{t-1}\lambda_{t-1} + 1 + \frac{\eta_{t-1}}{\eta_{t-2}}\gamma(1 - \alpha_{t-1}^{-1}), \forall t \geq 1. \tag{12}$$

needs to be always positive. Here $\alpha_0, \alpha_{-1}$ are free parameters. Different choice of $\alpha_0, \alpha_{-1}$ would lead to different trajectory for $\{\widetilde{\boldsymbol{\theta}}_t\}$, but the equality that $\widetilde{\boldsymbol{\theta}}_t = P_t\boldsymbol{\theta}_t$ is always satisfied. If the initial condition is given via $\boldsymbol{v}_0$, then it's also free to choose $\eta_{-1}, \boldsymbol{\theta}_{-1}$, as long as $\frac{\boldsymbol{\theta}_0 - \boldsymbol{\theta}_{-1}}{\eta_{-1}} = \boldsymbol{v}_0$.

*Proof of Theorem 2.13.* We will prove by induction. By assumption $S(t) : P_t\boldsymbol{\theta}_t = \widetilde{\boldsymbol{\theta}}_t$ for $t = -1, 0$. Now we will show that $S(t) \implies S(t + 1), \forall t \geq 0$.

$$\frac{\boldsymbol{\theta}_t - \boldsymbol{\theta}_{t-1}}{\eta_{t-1}} = \gamma\frac{\boldsymbol{\theta}_{t-1} - \boldsymbol{\theta}_{t-2}}{\eta_{t-2}} - \nabla_{\boldsymbol{\theta}}\left( (L(\boldsymbol{\theta}_{t-1}) + \frac{\lambda_{t-1}}{2}\|\boldsymbol{\theta}_{t-1}\|_2^2) \right)$$

$$\xrightarrow{\text{Take gradient}} \frac{\boldsymbol{\theta}_t - \boldsymbol{\theta}_{t-1}}{\eta_{t-1}} = \gamma\frac{\boldsymbol{\theta}_{t-1} - \boldsymbol{\theta}_{t-2}}{\eta_{t-2}} - \nabla_{\boldsymbol{\theta}}L(\boldsymbol{\theta}_{t-1}) + \lambda_{t-1}\boldsymbol{\theta}_{t-1}$$

$$\xrightarrow{\text{Scale Invariance}} \frac{\boldsymbol{\theta}_t - \boldsymbol{\theta}_{t-1}}{\eta_{t-1}} = \gamma\frac{\boldsymbol{\theta}_{t-1} - \boldsymbol{\theta}_{t-2}}{\eta_{t-2}} - P_{t-1}\nabla_{\boldsymbol{\theta}}L(\widetilde{\boldsymbol{\theta}}_{t-1}) + \lambda_{t-1}\boldsymbol{\theta}_{t-1}$$

$$\xrightarrow{\text{Rescaling}} \frac{P_t(\boldsymbol{\theta}_t - \boldsymbol{\theta}_{t-1})}{P_t P_{t-1}\eta_{t-1}} = \gamma\frac{P_{t-2}(\boldsymbol{\theta}_{t-1} - \boldsymbol{\theta}_{t-2})}{P_{t-1}P_{t-2}\eta_{t-2}} - \nabla_{\boldsymbol{\theta}}L(\widetilde{\boldsymbol{\theta}}_{t-1}) - \lambda_{t-1}\frac{\boldsymbol{\theta}_{t-1}}{P_{t-1}}$$

$$\xrightarrow{\text{Simplfying}} \frac{P_t\boldsymbol{\theta}_t - \alpha_{t-1}^{-1}\widetilde{\boldsymbol{\theta}}_{t-1}}{\widetilde{\eta}_{t-1}} = \gamma\frac{\alpha_{t-1}\widetilde{\boldsymbol{\theta}}_{t-1} - \widetilde{\boldsymbol{\theta}}_{t-2}}{\widetilde{\eta}_{t-2}} - \nabla_{\boldsymbol{\theta}}L(\widetilde{\boldsymbol{\theta}}_{t-1}) - \eta_{t-1}\lambda_{t-1}\frac{P_t\boldsymbol{\theta}_{t-1}}{\eta_{t-1}P_{t-1}P_t}$$

$$\xrightarrow{\text{Simplfying}} \frac{P_t\boldsymbol{\theta}_t - \alpha_t^{-1}\widetilde{\boldsymbol{\theta}}_{t-1}}{\widetilde{\eta}_{t-1}} = \gamma\frac{\alpha_{t-1}\widetilde{\boldsymbol{\theta}}_{t-1} - \widetilde{\boldsymbol{\theta}}_{t-2}}{\widetilde{\eta}_{t-2}} - \nabla_{\boldsymbol{\theta}}L(\widetilde{\boldsymbol{\theta}}_{t-1}) - \eta_{t-1}\lambda_{t-1}\frac{\alpha_t^{-1}\widetilde{\boldsymbol{\theta}}_{t-1}}{\widetilde{\eta}_{t-1}}$$

$$\xrightarrow{\text{Simplfying}} \frac{P_t\boldsymbol{\theta}_t - \alpha_t^{-1}(1 - \eta_{t-1}\lambda_{t-1})\widetilde{\boldsymbol{\theta}}_{t-1}}{\widetilde{\eta}_{t-1}} = \gamma\frac{\alpha_{t-1}\widetilde{\boldsymbol{\theta}}_{t-1} - \widetilde{\boldsymbol{\theta}}_{t-2}}{\widetilde{\eta}_{t-2}} - \nabla_{\boldsymbol{\theta}}L(\widetilde{\boldsymbol{\theta}}_{t-1})$$

To conclude that $P_t \boldsymbol{\theta}_t = \widetilde{\boldsymbol{\theta}}_t$, it suffices to show that the coefficients before $\widetilde{\boldsymbol{\theta}}_{t-1}$ is the same to that in (2). In other words, we need to show

$$\frac{-1 + \alpha_t^{-1}(1 - \eta_{t-1}\lambda_{t-1})}{\widetilde{\eta}_{t-1}} = \frac{\gamma(1 - \alpha_{t-1})}{\widetilde{\eta}_{t-2}},$$

which is equivalent to the definition of $\alpha_t$, Equation 12.

$\square$

**Lemma B.7** (Sufficient Conditions for positivity of $\alpha_t$). *Let* $\lambda_{max} = \max_t \lambda_t, \eta_{max} = \max_t \eta_t$. *Define* $z_{min}$ *is the larger root of the equation* $x^2 - (1 + \gamma - \lambda_{max}\eta_{max})x + \gamma = 0$. *To guarantee the existence of* $z_{max}$ *we also assume* $\eta_{max}\lambda_{max} \leq (1 - \sqrt{\gamma})^2$. *Then we have*

$$\forall \alpha_{-1}, \quad \alpha_0 = 1 \implies z_{min} \leq \alpha_t \leq 1, \forall t \geq 0 \tag{13}$$

*Proof.* We will prove the above theorem with a strengthened induction —

$$S(t) : \qquad \forall 0 \leq t' \leq t, \ z_{min} \leq \alpha_{t'} \leq 1 \bigwedge \frac{\alpha_{t'}^{-1} - 1}{\eta_{t'-1}} \leq \frac{z_{min}^{-1} - 1}{\eta_{max}}.$$

Since $\alpha_0 = 1$, $S(0)$ is obviously true. Now suppose $S(t)$ is true for some $t \in \mathbb{N}$, we will prove $S(t+1)$.

First, since $0 < \alpha_t \leq 1$, $\alpha_{t+1} = -\eta_t \lambda_t + 1 + \frac{\eta_t}{\eta_{t-1}}\gamma(1 - \alpha_t^{-1}) \leq 1$.

Again by Equation 12, we have

$$1 - \alpha_{t+1} = \eta_t \lambda_t + \frac{\alpha_t^{-1} - 1}{\eta_{t-1}}\eta_t\gamma = \eta_t\lambda_t + \frac{z_{min}^{-1} - 1}{\eta_{max}}\eta_t\gamma \leq \eta_t\lambda_t + (z_{min}^{-1} - 1)\gamma = 1 - z_{min},$$

which shows $\alpha_{t+1} \geq z_{min}$. Here the last step is by definition of $z_{min}$.

Because of $\alpha_{t+1} \geq z_{min}$, we have

$$\frac{\alpha_{t+1}^{-1} - 1}{\eta_t} \leq z_{min}^{-1}\frac{1 - \alpha_{t+1}}{\eta_t} \leq z_{min}^{-1}(\lambda_t + \frac{\alpha_t^{-1} - 1}{\eta_{t-1}}\gamma) \leq z_{min}^{-1}(\lambda_{max} + \frac{z_{min}^{-1} - 1}{\eta_{max}}\gamma) = z_{min}^{-1}\frac{1 - z_{min}}{\eta_{max}} = \frac{z_{min}^{-1} - 1}{\eta_{max}}.$$

$\square$

Now we are ready to give the formal statement about the closeness of Equation 9 and the reduced LR schedule by Theorem 2.13.

**Theorem B.8.** Given a Step Decay LR schedule with $\{T_I\}_{I=0}^{K-1}, \{\eta_I^*\}_{I=0}^{K-1}, \{\lambda_I^*\}_{I=0}^{K-1}$, the TEXP++ LR schedule in Theorem 2.13 is the following($\alpha_0 = \alpha_{-1} = 1, T_0 = 0$):

$$\alpha_t = \begin{cases} -\eta_I^*\lambda_I^* + 1 + \gamma(1 - \alpha_{t-1}^{-1}), \forall T_I + 2 \leq t \leq T_{I+1}, I \geq 0; \\ -\eta_I^*\lambda_I^* + 1 + \frac{\eta_I^*}{\eta_{I-1}^*}\gamma(1 - \alpha_{t-1}^{-1}), \forall t = T_I + 1, I \geq 0; \end{cases}$$

$$P_t = \prod_{i=-1}^{t} \alpha_t^{-1};$$

$$\hat{\eta}_t = P_t P_{t+1}\eta_t.$$

It's the same as the TEXP LR schedule($\{\widetilde{\eta}_t\}$) in Theorem 2.12 throughout each phase $I$, in the sense that

$$\left|\frac{\hat{\eta}_{t-1}}{\hat{\eta}_t} \Big/ \frac{\widetilde{\eta}_{t-1}}{\widetilde{\eta}_t} - 1\right| < 3\frac{\lambda_{max}\eta_{max}}{1 - \gamma}\left(\frac{\gamma}{z_{min}^2}\right)^{t-T_I-1} \leq 3\frac{\lambda_{max}\eta_{max}}{1 - \gamma}\left[\gamma(1 + \frac{\lambda_{max}\eta_{max}}{1 - \gamma})^2\right]^{(t-T_I-1)}, \quad \forall T_I + 1 \leq t \leq T_{I+1}.$$

where $z_{min}$ is the larger root of $x^2 - (1 + \gamma - \lambda_{max}\eta_{max})x + \gamma = 0$. In Appendix B, we show that $z_{min}^{-1} \leq 1 + \frac{\eta_{max}\lambda_{max}}{1-\gamma}$. When $\lambda_{max}\eta_{max}$ is small compared to $1 - \gamma$, which is usually the case in practice, one could approximate $z_{min}$ by 1. For example, when $\gamma = 0.9$, $\lambda_{max} = 0.0005$, $\eta_{max} = 0.1$, the above upper bound becomes

$$\left| \frac{\hat{\eta}_{t-1}}{\hat{\eta}_t} \Big/ \frac{\widetilde{\eta}_{t-1}}{\widetilde{\eta}_t} - 1 \right| \leq 0.0015 \times 0.9009^{t-T_I-1}.$$

*Proof of Theorem B.8.* Assuming $z_I^1$ and $z_I^2(z_I^1 \geq z_I^2)$ are the roots of Equation 1 with $\eta = \eta_I$ and $\lambda = \lambda_I$, we have $\gamma \leq z_{I'}^2 \leq \sqrt{\gamma} \leq z_{min} \leq z_I^1 \leq 1$, $\forall I, I' \in [K-1]$ by Lemma B.1.

We can rewrite the recursion in Theorem 2.13 as the following:

$$\alpha_t = -\eta_I \lambda_I + 1 + \gamma(1 - \alpha_{t-1}^{-1}) = -(z_I^1 + z_I^2) + z_I^1 z_I^2 \alpha_{t-1}^{-1}. \tag{14}$$

In other words, we have

$$\alpha_t - z_I^1 = \frac{z_I^2}{\alpha_{t-1}}(\alpha_{t-1} - z_I^1), t \geq 1. \tag{15}$$

By Lemma B.7, we have $\alpha_t \geq z_{min}$, $\forall t \geq 0$. Thus $|\frac{\alpha_t}{z_I^1} - 1| = \frac{z_2^I}{\alpha_{t-1}}|\frac{\alpha_{t-1}}{z_I^1} - 1| \leq \frac{\gamma}{z_{min}^2}|\frac{\alpha_{t-1}}{z_I^1} - 1| = \frac{\gamma}{z_{min}^2}|\frac{\alpha_{t-1}}{z_I^1} - 1| \leq \gamma(1 + \frac{\lambda\eta}{1-\gamma})^2|\frac{\alpha_{t-1}}{z_I^1}|$, which means $\alpha_t$ geometrically converges to its stable fixed point $z_I^1$. and $\frac{\widetilde{\eta}_{t-1}}{\widetilde{\eta}_t} = (z_I^1)^2$. Since that $z_{min} \leq \alpha_t \leq 1$, $z_{min} \leq z_I^1 \leq 1$, we have $|\frac{\alpha_{T_I}}{z_I^1} - 1| \leq \frac{1-z_{min}}{z_{min}} = \frac{\lambda_{max}\eta_{max}}{1-\gamma} \leq 1$, and thus $|\frac{\alpha_t}{z_I^1} - 1| \leq \frac{\lambda_{max}\eta_{max}}{1-\gamma}(\frac{\gamma}{z_{min}^2})^{t-T_I-1} \leq 1$, $\forall T_I + 1 \leq t \leq T_{I+1}$.

Note that $\alpha_I^* = z_I^1$, $\frac{\hat{\eta}_{t-1}}{\hat{\eta}_t} = \alpha_t \alpha_{t+1}$ By definition of TEXP and TEXP++, we have

$$\frac{\widetilde{\eta}_{t-1}}{\widetilde{\eta}_t} = \begin{cases} (z_{I-1}^1)^2 & \text{if } T_{I-1} + 1 \leq t \leq T_I - 1 \\ \frac{\eta_{I-1}^*}{\eta_I^*} z_I^1 z_{I-1}^1 & \text{if } t = T_I, I \geq 1 \end{cases} \tag{16}$$

$$\frac{\hat{\eta}_{t-1}}{\hat{\eta}_t} = \frac{\eta_{t-1}}{\eta_t}\alpha_{t+1}\alpha_t = \begin{cases} \alpha_{t+1}\alpha_t & \text{if } T_{I-1} + 1 \leq t \leq T_I - 1 \\ \frac{\eta_{I-1}^*}{\eta_I^*}\alpha_{T_I+1}\alpha_{T_I} & \text{if } t = T_I, I \geq 1 \end{cases} \tag{17}$$

Thus we have when $t = T_I$,

$$\left| \frac{\hat{\eta}_{t-1}}{\hat{\eta}_t} \Big/ \frac{\widetilde{\eta}_{t-1}}{\widetilde{\eta}_t} - 1 \right| \leq \left| \frac{\alpha_{T_I+1}}{z_I^1}\frac{\alpha_{T_I}}{z_{I-1}^1} - 1 \right| \leq \left| \frac{\alpha_{T_I+1}}{z_I^1} - 1 \right| + \left| \frac{\alpha_{T_I}}{z_{I-1}^1} - 1 \right| + \left| \frac{\alpha_{T_I+1}}{z_I^1} - 1 \right|\left| \frac{\alpha_{T_I}}{z_{I-1}^1} - 1 \right|$$

$$\leq 3\frac{\lambda_{max}\eta_{max}}{1-\gamma}.$$

When $T_I + 1 \leq t \leq T_{I+1}$, we have

$$\left| \frac{\hat{\eta}_{t-1}}{\hat{\eta}_t} \Big/ \frac{\widetilde{\eta}_{t-1}}{\widetilde{\eta}_t} - 1 \right| = \left| \frac{\alpha_{t+1}}{z_{I-1}^1}\frac{\alpha_t}{z_{I-1}^1} - 1 \right| \leq \left| \frac{\alpha_{t+1}}{z_{I-1}^1} - 1 \right| + \left| \frac{\alpha_t}{z_{I-1}^1} - 1 \right| + \left| \frac{\alpha_{t+1}}{z_{I-1}^1} - 1 \right|\left| \frac{\alpha_t}{z_{I-1}^1} - 1 \right|$$

$$\leq 3\frac{\lambda_{max}\eta_{max}}{1-\gamma}(\frac{\gamma}{z_{min}^2})^{t-T_I-1}.$$

Thus we conclude $\forall I \in [K-1], T_I + 1 \leq t \leq T_{I+1}$, we have

$$\left| \frac{\hat{\eta}_{t-1}}{\hat{\eta}_t} \Big/ \frac{\widetilde{\eta}_{t-1}}{\widetilde{\eta}_t} - 1 \right| \leq 3\frac{\lambda_{max}\eta_{max}}{1-\gamma}\left(\frac{\gamma}{z_{min}^2}\right)^{t-T_I-1} \leq 3\frac{\lambda_{max}\eta_{max}}{1-\gamma}\cdot\gamma^{t-T_I-1}(1+\frac{\lambda_{max}\eta_{max}}{1-\gamma})^{2(t-T_I-1)}.$$

$\square$

### B.6 Omitted Proofs in Section 3

We will use $\hat{\boldsymbol{w}}$ to denote $\frac{\boldsymbol{w}}{\|\boldsymbol{w}\|}$ and $\angle\boldsymbol{uw}$ to $\arccos(\hat{\boldsymbol{u}}^\top\hat{\boldsymbol{w}})$. Note that training error $\leq \frac{\varepsilon}{\pi}$ is equivalent to $\angle\boldsymbol{e}_1\boldsymbol{w}_t < \varepsilon$.

**Case 1: WD alone**  Since the objective is strongly convex, it has unique argmin $\boldsymbol{w}^*$. By symmetry, $\boldsymbol{w}^* = \beta\boldsymbol{e}_1$, for some $\beta > 0$. By KKT condition, we have

$$\lambda\beta = \mathop{\mathbb{E}}_{x_1\sim\mathcal{N}(0,1)}\left[\frac{|x_1|}{1 + \exp(\beta|x_1|)}\right] \leq \mathop{\mathbb{E}}_{x_1\sim\mathcal{N}(0,1)}[|x_1|] = \sqrt{\frac{2}{\pi}},$$

which implies $\|\boldsymbol{w}^*\| = O(\frac{1}{\lambda})$.

By Theorem 3.1 of Gower et al. (2019), for sufficiently large $t$, we have $\mathbb{E}\|\boldsymbol{w}_t - \boldsymbol{w}^*\|^2 = O(\frac{\eta}{B\lambda})$. Note that $\angle\boldsymbol{e}_1\boldsymbol{w}_t = \angle\boldsymbol{w}^*\boldsymbol{w}_t \leq 2\sin\angle\boldsymbol{w}^*\boldsymbol{w}_t \leq 2\frac{\|\boldsymbol{w}^*-\boldsymbol{w}^t\|}{\|\boldsymbol{w}^*\|}$, we have $\mathbb{E}(\angle\boldsymbol{e}_1\boldsymbol{w}_t)^2 = O(\frac{\eta\lambda}{B})$, so the expected error $= \mathbb{E}(\angle\boldsymbol{e}_1\boldsymbol{w}_t)/\pi \leq \sqrt{\mathbb{E}(\angle\boldsymbol{e}_1\boldsymbol{w}_t)^2}/\pi = O(\sqrt{\frac{\eta\lambda}{B}})$.

**Case 3: Both BN and WD**  We will need the following lemma when lower bounding the norm of the stochastic gradient.

**Lemma B.9** (Concentration of Chi-Square)**.** *Suppose* $X_1, \dots, X_k \overset{i.i.d.}{\sim} \mathcal{N}(0,1)$*, then*

$$Pr\left[\sum_{i=1}^{k} X_i^2 < k\beta\right] \leq \left(\beta e^{1-\beta}\right)^{\frac{k}{2}}. \tag{18}$$

*Proof.* This Chernoff-bound based proof is a special case of Dasgupta & Gupta (2003).

$$\begin{aligned}
Pr\left[\sum_{i=1}^{k} X_i^2 < k\beta\right] \leq \left(\beta e^{1-\beta}\right)^{\frac{k}{2}} &= Pr\left[\exp\left(kt\beta - t\sum_{i=1}^{k} X_i^2\right) \geq 1\right] \\
&\leq \mathbb{E}\left[\exp\left(kt\beta - t\sum_{i=1}^{k} X_i^2\right)\right] \text{ (Markov Inequality)} \\
&= e^{kt\beta}(1 + 2t)^{-\frac{k}{2}}.
\end{aligned} \tag{19}$$

The last equality uses the fact that $\mathbb{E}\left[tX_i^2\right] = \frac{1}{\sqrt{1-2t}}$ for $t < \frac{1}{2}$. The proof is completed by taking $t = \frac{1-\beta}{2\beta}$. $\qquad\square$

**Setting for Theorem B.6:**  Suppose WD factor is $\lambda$, LR is $\eta$, the width of the last layer is $m \geq 3$, Now the SGD updates have the form

$$\begin{aligned}
\boldsymbol{w}_{t+1} &= \boldsymbol{w}_t - \frac{\eta}{B}\sum_{b=1}^{B} \nabla\left(\ln(1 + \exp(-\boldsymbol{x}_{t,b}^\top \frac{\boldsymbol{w}_t}{\|\boldsymbol{w}_t\|}y_{t,b})) + \frac{\lambda}{2}\|\boldsymbol{w}_t\|^2\right) \\
&= (1 - \lambda\eta)\boldsymbol{w}_t - \frac{\eta}{B}\sum_{b=1}^{B} \frac{y_{t,b}}{1 + \exp(\boldsymbol{x}_{t,b}^\top \frac{\boldsymbol{w}_t}{\|\boldsymbol{w}_t\|}y_{t,b})} \frac{\Pi_{\boldsymbol{w}_t}^\perp \boldsymbol{x}_{t,b}}{\|\boldsymbol{w}_t\|},
\end{aligned}$$

where $\boldsymbol{x}_{t,b} \overset{i.i.d.}{\sim} \mathcal{N}(0, I_m), y_{t,b} = \text{sign}([x_{t,b}]_1)$, and $\Pi_{\boldsymbol{w}_t}^\perp = I - \frac{\boldsymbol{w}_t\boldsymbol{w}_t^\top}{\|\boldsymbol{w}_t\|^2}$.

*Proof of Theorem B.6.*

**Step 1:** Let $T_1 = \frac{1}{2(\eta\lambda - 2\varepsilon^2)} \ln \frac{64\|w_{T_0}\|^2 \varepsilon \sqrt{B}}{\eta\sqrt{m-2}}$, and $T_2 = 9\ln\frac{1}{\delta}$. Thus if we assume the training error is smaller than $\varepsilon$ from iteration $T_0$ to $T_0 + T_1 + T_2$, then by spherical triangle inequality, $\angle \boldsymbol{w}_t \boldsymbol{w}_{t'} \leq \angle \boldsymbol{e}_1 \boldsymbol{w}_{t'} + \angle \boldsymbol{e}_1 \boldsymbol{w}_t = 2\varepsilon$, for $T_0 \leq t, t' \leq T_0 + T_1 + T_2$.

Now let's define $\boldsymbol{w}_t' = (1 - \eta\lambda)\boldsymbol{w}_t$ and for any vector $\boldsymbol{w}$, and we have the following two relationships:

1. $\|\boldsymbol{w}_t'\| = (1 - \eta\lambda)\|\boldsymbol{w}\|$.

2. $\|\boldsymbol{w}_{t+1}\| \leq \frac{\|\boldsymbol{w}_t'\|}{\cos 2\varepsilon}$.

The second property is because by Lemma 1.3, $(\boldsymbol{w}_{t+1} - \boldsymbol{w}_t') \perp \boldsymbol{w}_t'$ and by assumption of small error, $\angle \boldsymbol{w}_{t+1} \boldsymbol{w}_t' \leq 2\varepsilon$.

Therefore

$$\frac{\|\boldsymbol{w}_{T_1+T_0}\|^2}{\|\boldsymbol{w}_{T_0}\|^2} \leq \left(\frac{1-\eta\lambda}{\cos 2\varepsilon}\right)^{2T_1} \leq \left(\frac{1-\eta\lambda}{1-2\varepsilon^2}\right)^{2T_1} \leq \left(1 - (\eta\lambda - 2\varepsilon^2)\right)^{2T_1} \leq e^{-2T_1(\eta\lambda - 2\varepsilon^2)} = \frac{\eta}{64\|w_{T_0}\|^2\varepsilon}\sqrt{\frac{m-2}{B}}.$$
$$(20)$$

In other word, $\|\boldsymbol{w}_{T_0+T_1}\|^2 \leq \frac{\eta}{64\varepsilon}\sqrt{\frac{m-2}{B}}$. Since $\|\boldsymbol{w}_{T_0+t}\|$ is monotone decreasing, $\|\boldsymbol{w}_{T_0+t}\|^2 \leq \frac{\eta}{64\varepsilon}\sqrt{\frac{m-2}{B}}$ holds for any $t = T_1, \ldots, T_1 + T_2$.

**Step 2:** We show that the norm of the stochastic gradient is lower bounded with constant probability. In other words, we want to show the norm of $\boldsymbol{\xi}_t = \sum_{b=1}^{B} \frac{y_{t,b}}{1+\exp(\boldsymbol{x}_{t,b}^\top \frac{\boldsymbol{w}_t}{\|\boldsymbol{w}_t\|}y_{t,b})} \frac{\Pi_{\boldsymbol{w}_t}^\perp \boldsymbol{x}_{t,b}}{\|\boldsymbol{w}_t\|}$ is lower bounded with high probability.

Let $\Pi_{\boldsymbol{w}_t, \boldsymbol{e}_1}^\perp$ be the projection matrix for the orthogonal space spanned by $\boldsymbol{w}_t$ and $\boldsymbol{e}_1$. W.L.O.G, we can assume the rank of $\Pi_{\boldsymbol{w}_t, \boldsymbol{e}_1}^\perp$ is 2. In case $\boldsymbol{w}_t = \boldsymbol{e}_1$, we just exclude a random direction to make $\Pi_{\boldsymbol{w}_t, \boldsymbol{e}_1}^\perp$ rank 2. Now we have $\Pi_{\boldsymbol{w}_t, \boldsymbol{e}_1}^\perp \boldsymbol{x}_{t,b}$ are still i.i.d. multivariate gaussian random variables, for $b = 1, \ldots, B$, and moreover, $\Pi_{\boldsymbol{w}_t, \boldsymbol{e}_1}^\perp \boldsymbol{x}_{t,b}$ is independent to $\frac{y_{t,b}}{1+\exp(\boldsymbol{x}_{t,b}^\top \frac{\boldsymbol{w}_t}{\|\boldsymbol{w}_t\|}y_{t,b})}$. When $m \geq 3$, we can lower bound $\|\boldsymbol{\xi}_t\|$ by dealing with $\|\Pi_{\boldsymbol{w}_t, \boldsymbol{e}_1}^\perp \boldsymbol{\xi}_t\|$.

It's not hard to show that conditioned on $\{\boldsymbol{x}_{t,b}^\top \frac{\boldsymbol{w}_t}{\|\boldsymbol{w}_t\|}, [\boldsymbol{x}_{t,b}]_1\}_{b=1}^{B}$,

$$\sum_{b=1}^{B} \frac{y_{t,b}}{1 + \exp(\boldsymbol{x}_{t,b}^\top \frac{\boldsymbol{w}_t}{\|\boldsymbol{w}_t\|}y_{t,b})}\Pi_{\boldsymbol{w}_t}^\perp \boldsymbol{x}_{t,b} \stackrel{d}{=} \sqrt{\sum_{b=1}^{B}\left(\frac{y_{t,b}}{1 + \exp(\boldsymbol{x}_{t,b}^\top \frac{\boldsymbol{w}_t}{\|\boldsymbol{w}_t\|}y_{t,b})}\right)^2}\Pi_{\boldsymbol{w}_t, \boldsymbol{e}_1}^\perp \boldsymbol{x}, \quad (21)$$

where $\boldsymbol{x} \sim \mathcal{N}(\boldsymbol{0}, I_m)$. We further note that $\|\Pi_{\boldsymbol{w}_t, \boldsymbol{e}_1}^\perp \boldsymbol{x}\|^2 \sim \chi^2(m-2)$. By Lemma B.9,

$$\Pr\left[\|\Pi_{\boldsymbol{w}_t, \boldsymbol{e}_1}^\perp \boldsymbol{x}_t\|^2 \geq \frac{m-2}{8}\right] \geq 1 - (\frac{1}{8e^{\frac{7}{8}}})^{\frac{m-2}{2}} \geq 1 - (\frac{1}{8e^{\frac{7}{8}}})^{\frac{1}{2}} \geq \frac{1}{3}. \quad (22)$$

Now we will give a high probability lower bound for $\sum_{b=1}^{B}\left(\frac{y_{t,b}}{1+\exp(\boldsymbol{x}_{t,b}^\top \frac{\boldsymbol{w}_t}{\|\boldsymbol{w}_t\|}y_{t,b})}\right)^2$. Note that $\boldsymbol{x}_t^\top \frac{\boldsymbol{w}_t}{\|\boldsymbol{w}_t\|} \sim \mathcal{N}(0, 1)$, we have

$$\Pr\left[|\boldsymbol{x}_{t,b}^\top \frac{\boldsymbol{w}_t}{\|\boldsymbol{w}_t\|}| < 1\right] \geq \frac{1}{2}, \quad (23)$$

which implies the following, where $A_{t,b}$ is defined as $\mathbb{1}\left[|\boldsymbol{x}_{t,b}^\top \frac{\boldsymbol{w}_t}{\|\boldsymbol{w}_t\|}| < 1 \geq \frac{1}{2}\right]$:

$$\mathbb{E}\, A_{t,b} = \Pr\left[\|\frac{y_{t,b}}{1 + \exp(\boldsymbol{x}_{t,b}^\top \frac{\boldsymbol{w}_t}{\|\boldsymbol{w}_t\|} y_t)}\| \geq \frac{1}{1+e}\right] \geq \frac{1}{2}. \tag{24}$$

Note that $\sum_{b=1}^{B} A_{t,b} \leq B$, and $\mathbb{E}\sum_{b=1}^{B} A_{t,b} \geq \frac{B}{2}$, we have $\Pr\left[\sum_{b=1}^{B} A_{t,b} < \frac{B}{4}\right] \leq \frac{2}{3}$. Thus,

$$\Pr\left[\sum_{b=1}^{B}\left(\frac{y_{t,b}}{1 + \exp(\boldsymbol{x}_{t,b}^\top \frac{\boldsymbol{w}_t}{\|\boldsymbol{w}_t\|} y_{t,b})}\right)^2 \geq \frac{B}{4(1+e)^2}\right] \geq \Pr\left[\sum_{b=1}^{B} A_{t,b} \geq \frac{B}{4}\right] \geq \frac{1}{3}. \tag{25}$$

Thus w.p. at least $\frac{1}{9}$, equation 25 and equation 22 happen together, which implies

$$\|\frac{\eta}{B}\sum_{b=1}^{B}\nabla\ln(1+\exp(-\boldsymbol{x}_{t,b}^\top \frac{\boldsymbol{w}_t}{\|\boldsymbol{w}_t\|} y_{t,b}))\| = \|\frac{\eta}{B}\sum_{b=1}^{B}\frac{y_{t,b}}{1 + \exp(\boldsymbol{x}_{t,b}^\top \frac{\boldsymbol{w}_t}{\|\boldsymbol{w}_t\|} y_t)}\frac{\Pi_{\boldsymbol{w}_t}^\perp \boldsymbol{x}_{t,b}}{\|\boldsymbol{w}_t\|}\| \geq \frac{\eta}{1+e}\frac{\sqrt{m-2}}{8\|\boldsymbol{w}_t\|} \geq \frac{\eta}{32\|\boldsymbol{w}_t\|}\sqrt{\frac{m-2}{B}} \tag{26}$$

**Step 3**. To stay in the cone $\{\boldsymbol{w}|\angle \boldsymbol{w}\boldsymbol{e}_1 \leq \varepsilon\}$, the SGD update $\|\boldsymbol{w}_{t+1} - \boldsymbol{w}_t'\| = \|\frac{\eta}{B}\sum_{b=1}^{B}\nabla\ln(1 + \exp(-\boldsymbol{x}_{t,b}^\top \frac{\boldsymbol{w}_t}{\|\boldsymbol{w}_t\|} y_{t,b}))\|$ has to be smaller than $\|\boldsymbol{w}_t\|\sin 2\varepsilon$ for any $t = T_0 + T_1, \ldots, T_0 + T_1 + T_2$. However, step 1 and 2 together show that $\|\nabla\ln(1 + \exp(-\boldsymbol{x}_t^\top \frac{\boldsymbol{w}_t}{\|\boldsymbol{w}_t\|} y_t))\| \geq 2\|\boldsymbol{w}_t\|\varepsilon$ w.p. $\frac{1}{9}$ per iteration. Thus the probability that $\boldsymbol{w}_t$ always stays in the cone for every $t = T_0 + T_1, \ldots, T_0 + T_1 + T_2$ is less than $\left(\frac{8}{9}\right)^{T_2} \leq \delta$. $\square$

**Remark B.10.** It's interesting that the only property of the global minimum we use is that the if both $\boldsymbol{w}_t$, $\boldsymbol{w}_{t+1}$ are $\epsilon$ optimal, then the angle between $\boldsymbol{w}_t$ and $\boldsymbol{w}_{t+1}$ is at most $2\epsilon$. Thus we indeed have proved a stronger statement: *At least once in every* $\frac{1}{2(\eta\lambda - 2\varepsilon^2)}\ln\frac{64\|w_{T_0}\|^2\varepsilon\sqrt{B}}{\eta\sqrt{m-2}} + 9\ln\frac{1}{\delta}$ *iterations, the angle between $\boldsymbol{w}_t$ and $\boldsymbol{w}_{t+1}$ will be larger than $2\epsilon$.* In other words, if the the amount of the update stablizes in terms of angle, then this angle must be larger than $\sqrt{2\eta\lambda}$ for this simple model.

## B.7  OMITTED PROOFS IN SECTION A

**Lemma B.11.** *Suppose loss $L$ is scale invariant, then $L$ is non-convex in the following two sense:*

1. *The domain is non-convex: scale invariant loss can't be defined at origin;*

2. *There exists no ball containing origin such that the loss is locally convex, unless the loss is constant function.*

*Proof.* Suppose $L(\boldsymbol{\theta}^*) = \sup_{\boldsymbol{\theta}\in B} L(\boldsymbol{\theta})$. W.L.O.G, we assume $\|\boldsymbol{\theta}^*\| < 1$. By convexity, every line segment passing $\boldsymbol{\theta}^*$ must have constant loss, which implies the loss is constant over set $\mathcal{B} - \{c\frac{\boldsymbol{\theta}^*}{\|\boldsymbol{\theta}^*\|} \mid -1 \leq c \leq 0\}$. Applying the above argument on any other maximum point $\boldsymbol{\theta}'$ implies the loss is constant over $\mathcal{B} - \{\mathbf{0}\}$. $\square$

**Theorem B.12.** *Suppose the momentum factor $\gamma = 0$, LR $\eta_t = \eta$ is constant, and the loss function $L$ is lower bounded. If $\exists c > 0$ and $T \geq 0$ such that $\forall t \geq T$, $f(\boldsymbol{\theta}_{t+1}) - f(\boldsymbol{\theta}_t) \leq -c\eta\|\nabla L(\boldsymbol{\theta}_t)\|^2$, then $\lim_{t\to\infty}\|\boldsymbol{\theta}_t\| = 0$.*

*Proof in Item 3.* By Lemma 1.3 and the update rule of GD with WD, we have

$$\|\boldsymbol{\theta}_t\|^2 = \|(1 - \lambda\eta)\|\boldsymbol{\theta}\|_{t-1} + \eta\nabla L(\boldsymbol{\theta}_{t-1})\|^2 = (1 - \lambda\eta)^2\|\boldsymbol{\theta}_{t-1}\|^2 + \eta^2\|\nabla L(\boldsymbol{\theta}_{t-1})\|^2,$$

which implies

$$\|\boldsymbol{\theta}_t\|^2 = \sum_{i=T}^{t-1}(1 - \lambda\eta)^{2(t-i-1)}\eta^2\|\nabla L(\boldsymbol{\theta}_{t-1})\|^2 + (1 - \lambda\eta)^{2(t-T)}\|\boldsymbol{\theta}_T\|^2.$$

Thus for any $T' > T$,

$$\sum_{t=T}^{T'} \|\boldsymbol{\theta}_t\|^2 \leq \frac{1}{1-(1-\lambda\eta)^2} \left( \sum_{t=T}^{T'-1} \|\nabla L(\boldsymbol{\theta}_t)\|^2 + \|\boldsymbol{\theta}_T\|^2 \right) \leq \frac{1}{\lambda\eta} \left( \sum_{t=T}^{T'-1} \|\nabla L(\boldsymbol{\theta}_t)\|^2 + \|\boldsymbol{\theta}_T\|^2 \right).$$

Note that by assumption we have $\sum_{t=T}^{T'-1} \|\nabla L(\boldsymbol{\theta}_t)\|^2 = \frac{1}{c\eta} f(\boldsymbol{\theta}_T) - f(\boldsymbol{\theta}_{T'})$.

As a conclusion, we have $\sum_{t=T}^{\infty} \|\boldsymbol{\theta}_t\|^2 \leq \frac{f(\boldsymbol{\theta}_T) - \min_{\boldsymbol{\theta}} f(\boldsymbol{\theta})}{c\eta^2\lambda} + \frac{\|\boldsymbol{\theta}_T\|^2}{\lambda\eta}$, which implies $\lim_{t\to\infty} \|\boldsymbol{\theta}_t\|^2 = 0$. $\qquad\square$

## C  OTHER RESULTS

Now we rigorously analyze norm growth in this algorithm. This greatly extends previous analyses of effect of normalization schemes (Wu et al., 2018; Arora et al., 2018) for vanilla SGD.

**Theorem C.1.** Under the update rule 1.2 with $\lambda_t = 0$, the norm of scale invariant parameter $\boldsymbol{\theta}_t$ satisfies the following property:

- Almost Monotone Increasing: $\|\boldsymbol{\theta}_{t+1}\|^2 - \|\boldsymbol{\theta}_t\|^2 \geq -\gamma^{t+1} \frac{\eta_t}{\eta_0} (\|\boldsymbol{\theta}_0\|^2 - \|\boldsymbol{\theta}_{-1}\|^2)$.

- Assuming $\eta_t = \eta$ is a constant, then

$$\|\boldsymbol{\theta}_{t+1}\|^2 = \sum_{i=0}^{t} \frac{1-\gamma^{t-i+1}}{1-\gamma} \left( \|\boldsymbol{\theta}_i - \boldsymbol{\theta}_{i+1}\|^2 + \gamma\|\boldsymbol{\theta}_{i-1} - \boldsymbol{\theta}_i\|^2 \right) - \gamma\frac{1-\gamma^{t+1}}{1-\gamma}(\|\boldsymbol{\theta}_0\|^2 - \|\boldsymbol{\theta}_{-1}\|^2)$$

*Proof.* Let's use $R_t, D_t, C_t$ to denote $\|\boldsymbol{\theta}_t\|^2, \|\boldsymbol{\theta}_{t+1} - \boldsymbol{\theta}_t\|^2, \boldsymbol{\theta}_t^\top(\boldsymbol{\theta}_{t+1} - \boldsymbol{\theta}_t)$ respectively.

The only property we will use about loss is $\nabla_{\boldsymbol{\theta}} L_t^\top \boldsymbol{\theta}_t = 0$.

Expanding the square of $\|\boldsymbol{\theta}_{t+1}\|^2 = \|(\boldsymbol{\theta}_{t+1} - \boldsymbol{\theta}_t) + \boldsymbol{\theta}_t\|^2$, we have

$$\forall t \geq -1 \quad S(t): \quad R_{t+1} - R_t = D_t + 2C_t.$$

We also have

$$\frac{C_t}{\eta_t} = \boldsymbol{\theta}_t^\top \frac{\boldsymbol{\theta}_{t+1} - \boldsymbol{\theta}_t}{\eta_t} = \boldsymbol{\theta}_t^\top (\gamma \frac{\boldsymbol{\theta}_t - \boldsymbol{\theta}_{t-1}}{\eta_{t-1}} - \lambda_t \boldsymbol{\theta}_t) = \frac{\gamma}{\eta_{t-1}}(D_t + C_{t-1}) - \lambda_t R_t,$$

namely,

$$\forall t \geq 0 \quad P(t): \quad \frac{C_t}{\eta_t} - \frac{\gamma D_t}{\eta_{t-1}} = \frac{\gamma}{\eta_{t-1}} C_{t-1} - \lambda_t R_t.$$

Simplify $\frac{S(t)}{\eta_t} - \frac{\gamma S(t-1)}{\eta_{t-1}} + P(t)$, we have

$$\frac{R_{t+1} - R_t}{\eta_t} - \gamma \frac{R_t - R_{t-1}}{\eta_{t-1}} = \frac{D_t}{\eta_t} + \gamma \frac{D_{t-1}}{\eta_{t-1}} - 2\lambda_t R_t. \tag{27}$$

When $\lambda_t = 0$, we have

$$\frac{R_{t+1} - R_t}{\eta_t} = \gamma^{t+1} \frac{R_0 - R_{-1}}{\eta_{-1}} + \sum_{i=0}^{t} \gamma^{t-i} (\frac{D_i}{\eta_i} + \gamma\frac{D_{i-1}}{\eta_{i-1}}) \geq \gamma^{t+1} \frac{R_0 - R_{-1}}{\eta_0}.$$

Further if $\eta_t = \eta$ is a constant, we have

$$R_{t+1} = R_0 + \sum_{i=0}^{t} \frac{1-\gamma^{t-i+1}}{1-\gamma}(D_i + \gamma D_{i-1}) - \gamma\frac{1-\gamma^{t+1}}{1-\gamma}(R_0 - R_{-1}),$$

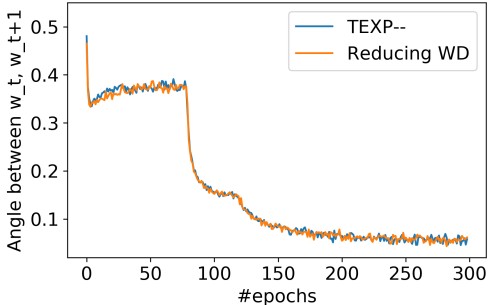 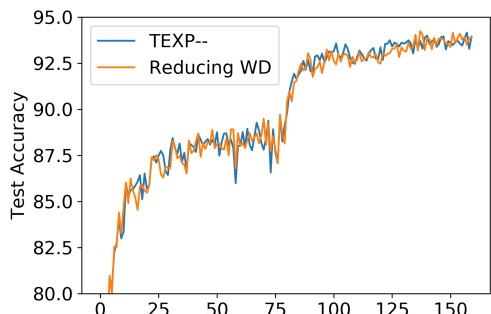

**Figure 6:** The orange line corresponds to PreResNet32 trained with constant LR and WD divided by 10 at epoch 80 and 120. The blue line is TEXP-- corresponding to Step Decay schedule which divides LR by 10 at epoch 80 and 120. They have similar trajectories and performances by a similar argument to Theorem 2.12.(See Theorem B.2 and its proof in Appendix B)

which covers the result without momentum in (Arora et al., 2019) as a special case:

$$R_{t+1} = R_0 + \sum_{i=0}^{t} D_i.$$

$\square$

For general deep nets, we have the following result, suggesting that the mean square of the update are constant compared to the mean square of the norm. The constant is mainly determined by $\eta\lambda$, explaining why the usage of weight decay prevents the parameters to converge in direction. [1]

**Theorem C.2.** For SGD with constant LR $\eta$, weight decay $\lambda$ and momentum $\gamma$, when the limits $R_\infty = \lim_{T\to\infty} \frac{1}{T} \sum_{t=0}^{T-1} \|w_t\|^2$, $D_\infty = \lim_{T\to\infty} \frac{1}{T} \sum_{t=0}^{T-1} \|w_{t+1} - w_t\|^2$ exist, we have

$$D_\infty = \frac{2\eta\lambda}{1+\gamma} R_\infty.$$

*Proof of Theorem C.2.* Take average of Equation 27 over $t$, when the limits $R_\infty = \lim_{T\to\infty} \frac{1}{T} \sum_{t=0}^{T-1} \|w_t\|^2$, $D_\infty = \lim_{T\to\infty} \frac{1}{T} \sum_{t=0}^{T-1} \|w_{t+1} - w_t\|^2$ exists, we have

$$\frac{1+\gamma}{\eta} D_\infty = 2\lambda R_\infty.$$

$\square$

# D    ADDITIONAL EXPERIMENTAL FIGURES

# E    SCALE INVARIANCE IN MODERN NETWORK ARCHITECTURES

In this section, we will discuss how Normalization layers make the output of the network scale-invariant to its parameters. Viewing a neural network as a DAG, we give a sufficient condition for the scale invariance which could be checked easily by topological order, and apply this on several standard network architectures such as Fully Connected(FC) Networks, Plain CNN, ResNet(He et al., 2016a), and PreResNet(He et al., 2016b). For simplicity, we restrict our discussions among networks with ReLU activation only. Throughout this section, we assume the linear layers and the bias after last normalization layer are fixed to its random initialization, which doesn't harm the performance of the network empirically(Hoffer et al., 2018b).

---

[1](Page) had a similar argument for this phenomenon by connecting this to the LARS(You et al., 2017), though it's not rigorous in the way it deals with momentum and equilibrium of norm.

### E.1 NOTATIONS

**Definition E.1** (Degree of Homogeneity). Suppose $k$ is an integer and $\boldsymbol{\theta}$ is all the parameters of the network, then $f$ is said to be *homogeneous of degree $k$, or $k$-homogeneous*, if $\forall c > 0$, $f(c\boldsymbol{\theta}) = c^k f(\boldsymbol{\theta})$. The output of $f$ can be multi-dimensional. Specifically, scale invariance means degree of homogeneity is 0.

Suppose the network only contains following modules, and we list the degree of homogeneity of these basic modules, given the degree of homogeneity of its input.

- (I) Input
- (L) Linear Layer, e.g. Convolutional Layer or Fully Connected Layer
- (B) Bias Layer(Adding Trainable Bias to the output of the previous layer)
- (+) Addition Layer (adding the outputs of two layers with the same dimension[2].)
- (N) Normalization Layer without affine transformation(including BN, GN, LN, IN etc.)
- (NA) Normalization Layer with affine transformation

| Module | I | L | B | + | N | NA |
|--------|---|-----|---|-------|---|----|
| Input | - | x | 1 | (x,x) | x | x |
| Output | 0 | x+1 | 1 | x | 0 | 1 |

**Table 1:** Table showing how degree of homogeneity of the output of basic modules depends on the degree of homogeneity of the input. For the row of the **Input** , entry '-' means the input of the network (I) doesn't have any extra input, entry '1' of Bias Layer means if the input is 1-homogeneous then the output is 1- homogeneous. '$(x, x)$' for '+' means if the inputs of Addition Layer have the same degree of homogeneity, the output has the same degree of homogeneity. ReLU, Pooling( and other fixed linear maps) are ignored because they keep the degree of homogeneity and can be omitted when creating the DAG in Theorem E.3.

**Remark E.2.** For the purpose of deciding the degree of homogeneity of a network, there's no difference among convolutional layers, fully connected layer and the diagonal linear layer in the affine transformation of Normalization layer, since they're all linear and the degree of homogeneity is increased by 1 after applying them.

On the other hand, BN and IN has some benefit which GN and LN doesn't have, namely the bias term (per channel) immediately before BN or IN has zero effect on the network output and thus can be removed. (See Figure 15)

We also demonstrate the homogeneity of the output of the modules via the following figures, which will be reused to later to define network architectures.

**Theorem E.3.** For a network only consisting of modules defined above and ReLU activation, we can view it as a *Directed acyclic graph* and check its scale invariance by the following algorithm.

---

**Input** : DAG $G = (V, E)$ translated from a neural network; the module type of each node $v_i \in V$.

1 **for** $v$ *in topological order of $G$* **do**
2      Compute the degree of homogeneity of $v$ using Table 1;
3      **if** $v$ *is not homogeneous* **then**
4          **return** *False*;
5 **if** $v_{ouptut}$ *is 0-homogeneous* **then**
6      **return** *True*;
7 **else**
8      **return** *False*.

---

[2] Addition Layer(+) is mainly used in ResNet and other similar architectures. In this section, we also use it as an alternative definition of Bias Layer(B). See Figure 7

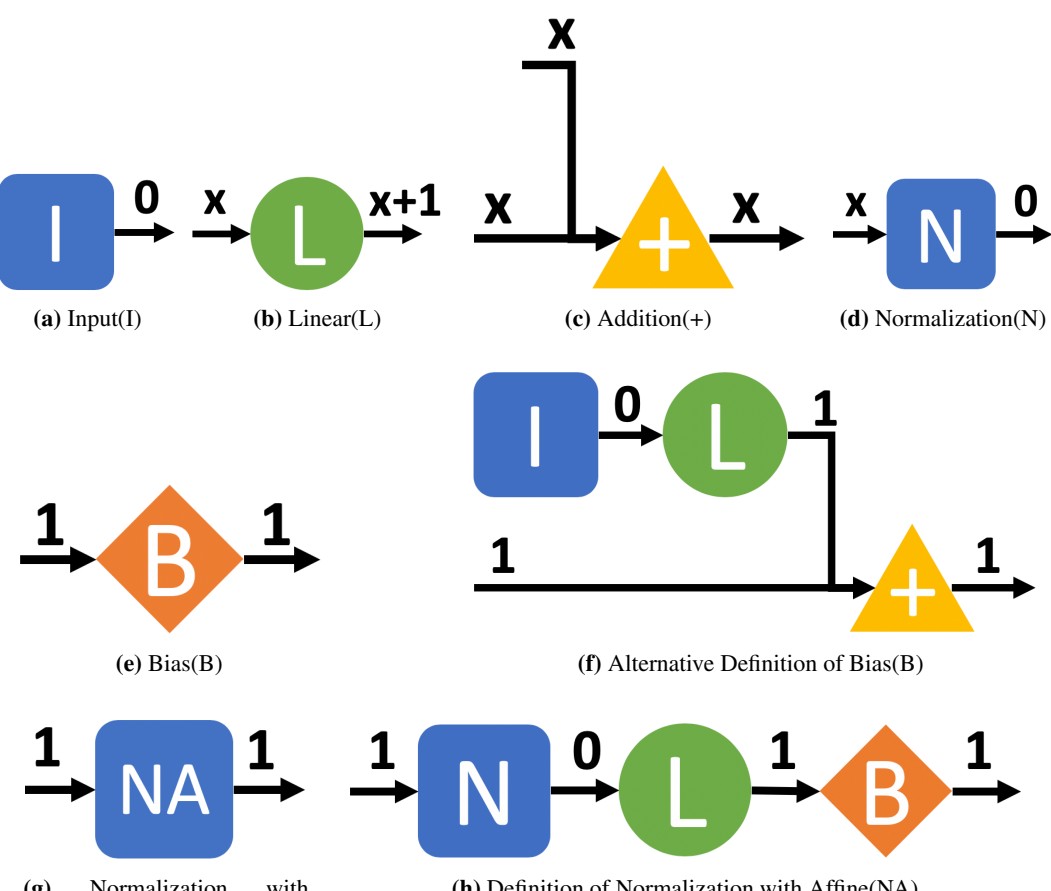

Figure 7: Degree of homogeneity of the output of basic modules given degree of homogeneity of the input.

### E.2 Networks without Affine Transformation and Bias

We start with the simple cases where all bias term(including that of linear layer and normalization layer) and the scaling term of normalization layer are fixed to be 0 and 1 element-wise respectively, which means the bias and the scaling could be dropped from the network structure. We empirically find this doesn't affect the performance of network in a noticeable way. We will discuss the full case in the next subsection.

**Plain CNN/FC networks:** See Figure 8.

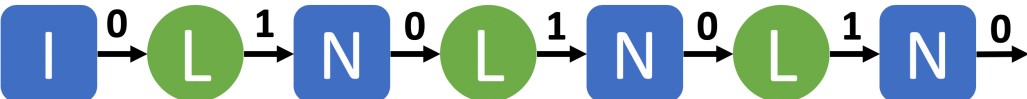

Figure 8: Degree of homogeneity for all modules in vanilla CNNs/FC networks.

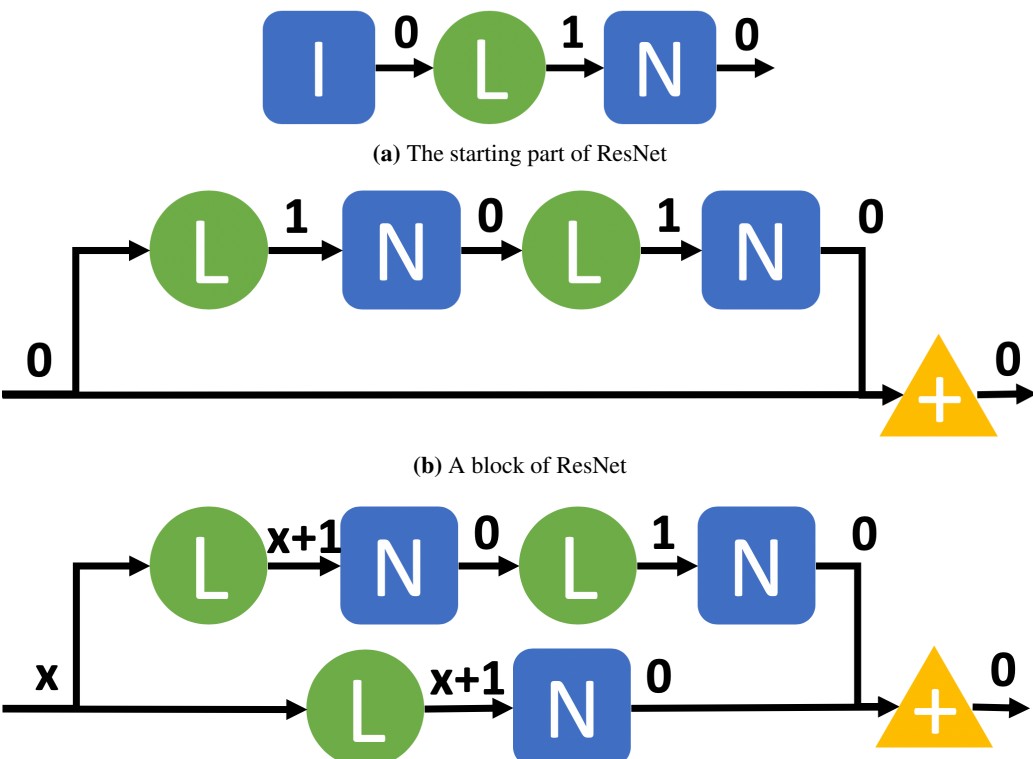

Figure 9: An example of the full network structure of ResNet/PreResNet represented by composite modules defined in Figure 10,11,13,14, where 'S' denotes the starting part of the network, 'Block' denotes a normal block with residual link, 'D-Block' denotes the block with downsampling, and 'N' denotes the normalization layer defined previously. Integer $x \in \{0, 1, 2\}$ depends on the type of network. See details in Figure 10,11,13,14.

**ResNet:** See Figure 10. To ensure the scaling invariance, we add an additional normalizaiton layer in the shortcut after downsampling. This implementation is sometimes used in practice and doesn't affect the performance in a noticeable way.

**(a)** The starting part of ResNet

**(b)** A block of ResNet

**(c)** A block of ResNet with downsampling

Figure 10: Degree of homogeneity for all modules in ResNet without affine transformation in normalization layer. The last normalization layer is omitted.

**Preactivation ResNet:**   See Figure 11. Preactivation means to change the order between convolutional layer and normalization layer. For similar reason, we add an additional normalizaiton layer in the shortcut before downsampling.

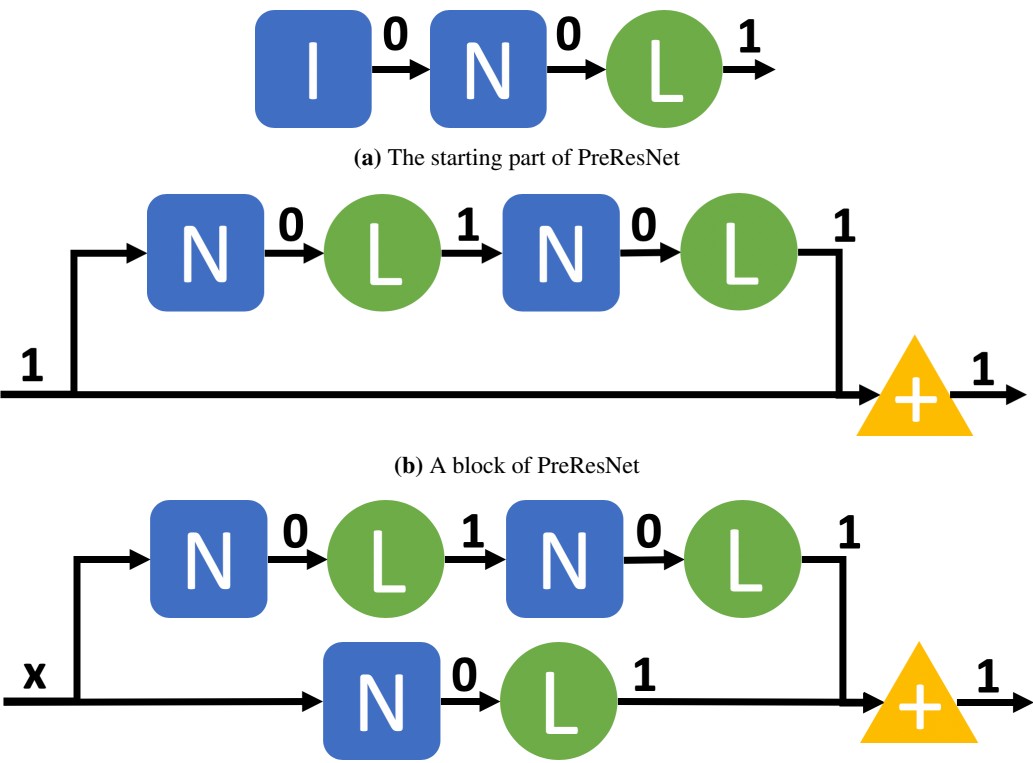

(a) The starting part of PreResNet

(b) A block of PreResNet

(c) A block of PreResNet with downsampling

**Figure 11:** Degree of homogeneity for all modules in ResNet without affine transformation in normalization layer. The last normalization layer is omitted.

### E.3    NETWORKS WITH AFFINE TRANSFORMATION

Now we discuss the full case where the affine transformation part of normalization layer is trainable. Due to the reason that the bias of linear layer (before BN) has 0 gradient as we mentioned in E.2, the bias term is usually dropped from network architecture in practice to save memory and accelerate training( even with other normalization methods)(See PyTorch Implementation (Paszke et al., 2017)). However, when LN or GN is used, and the bias term of linear layer is trainable, the network could be scale variant (See Figure 15).

**Plain CNN/FC networks:**    See Figure 12.

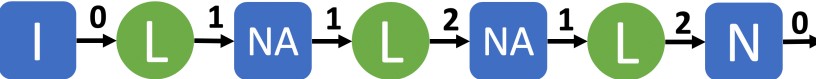

**Figure 12:** Degree of homogeneity for all modules in vanilla CNNs/FC networks.

**ResNet:**    See Figure 13. To ensure the scaling invariance, we add an additional normalizaiton layer in the shortcut after downsampling. This implementation is sometimes used in practice and doesn't affect the performance in a noticeable way.

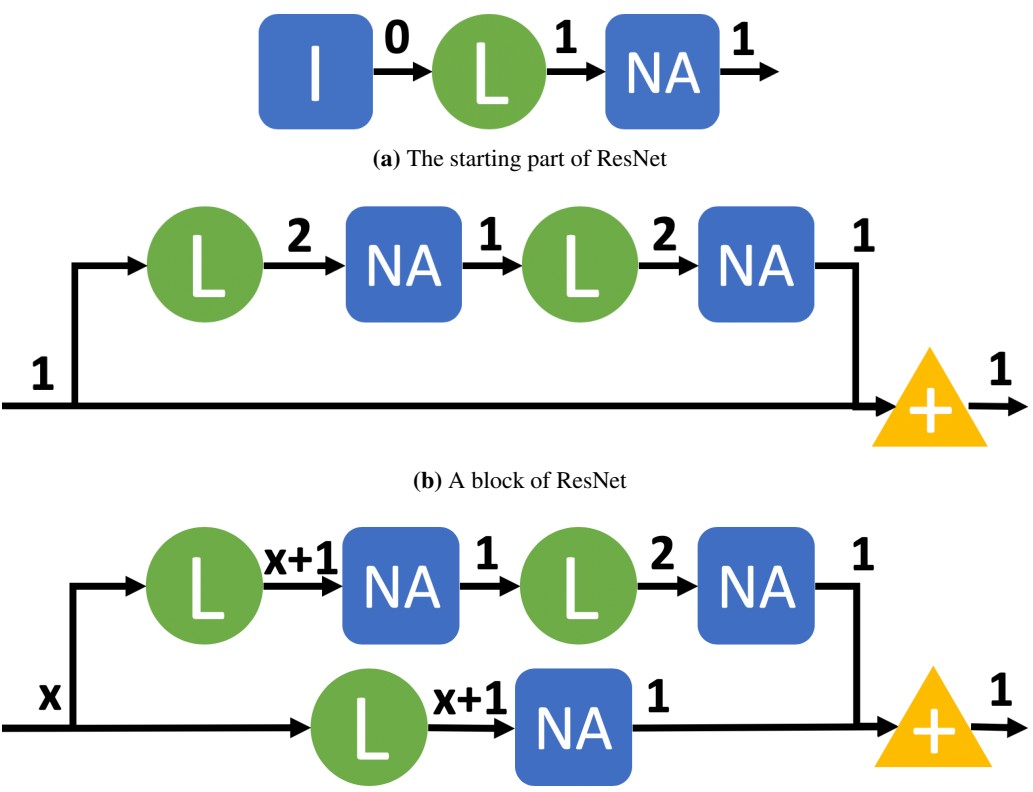

**(a)** The starting part of ResNet

**(b)** A block of ResNet

**(c)** A block of ResNet with downsampling

**Figure 13:** Degree of homogeneity for all modules in ResNet with trainable affine transformation. The last normalization layer is omitted.

**Preactivation ResNet:**    See Figure 14. Preactivation means to change the order between convolutional layer and normalization layer. For similar reason, we add an additional normalizaiton layer in the shortcut before downsampling.

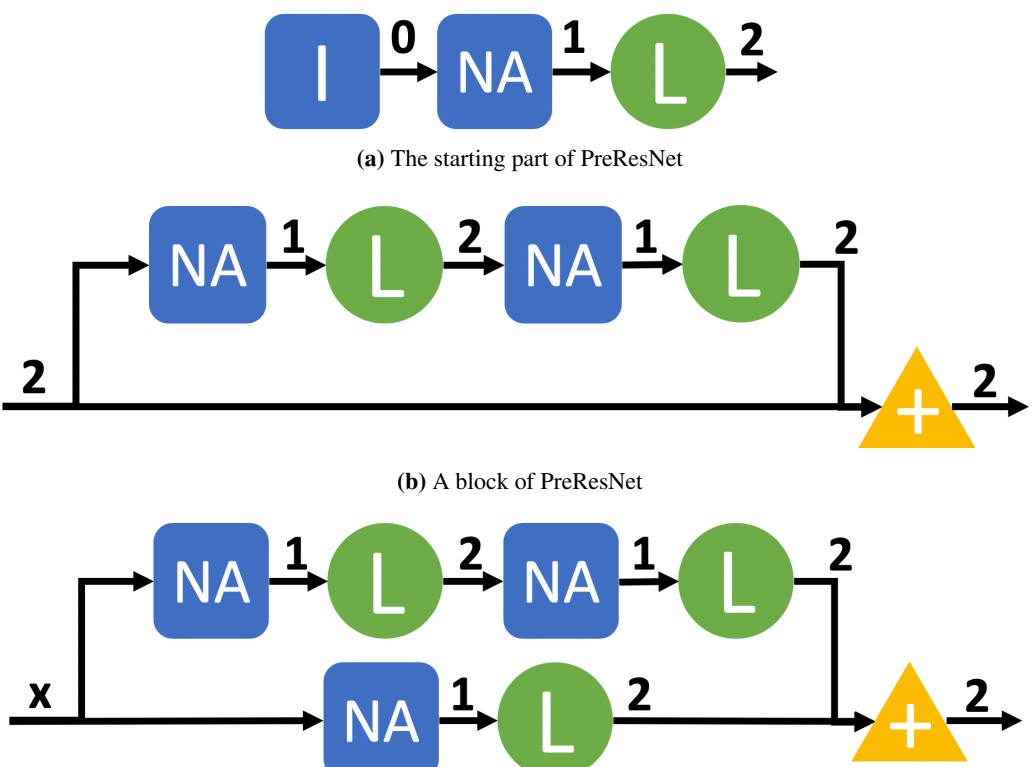

**(a)** The starting part of PreResNet

**(b)** A block of PreResNet

**(c)** A block of PreResNet with downsampling

**Figure 14:** Degree of homogeneity for all modules in PreResNet with trainable affine transformation. The last normalization layer is omitted.

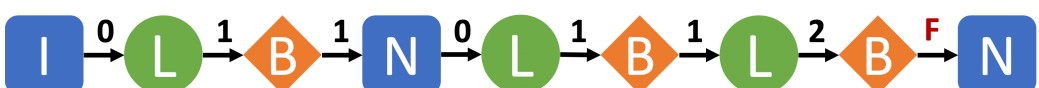

**Figure 15:** The network can be not scale variant if the GN or IN is used and the bias of linear layer is trainable. The red 'F' means the Algorithm 1 will return **False** here.

