# OpenReview forum: "An Exponential Learning Rate Schedule for Deep Learning"
_ICLR.cc/2020/Conference — Accept (Spotlight)_

### Official Review · AnonReviewer1 · 2019-10-23
**Official Blind Review #1**

**Rating:** 6

**Review:**

%%% Update to the review %%%
Thanks for your clarification and the revision - the paper looks good! With regards to your comment on accelerating hyper-parameter search, note that there are fairly subtle issues owing to the use of SGD - refer to a recent work of Ge et al "The Step Decay Schedule: A Near Optimal, Geometrically Decaying Learning Rate Procedure For Least Squares" (2019).
%%%

The paper makes an interesting observation connecting the use of weight decay + normalization to training the same network (without regularization) using an exponentially increasing learning rate schedule, under an assumption of scale invariance that is satisfied by normalization techniques including batch norm, layer norm and other variants. An example is also provided where the joint use of batch norm and weight decay can lead to non-convergence to a global minimum whereas the use of one (without the other) converges to a global minimum - serving to indicate various interdependencies between the hyper-parameters that one needs to be careful about.

While the connection of scale invariant models to novel schemes of learning rates is interesting (and novel), the paper will benefit quite a bit in its contributions through attempting a convergence analysis towards a stationary point even for solving a routine smooth non-convex stochastic optimization problem. Owing to the equivalence described in the paper, this enables us to understand the behavior of the combination of batch norm (or some scale invariance property) + weight decay + momentum (+ step decay of the learning rate), which, to my knowledge isn’t present in the literature of non-convex optimization.

The paper is reasonably written (the proof of the main claim is fairly easy to follow), but needs to be carefully read through because I see typos and ill-formed sentences that should be rectified - e.g. see point 3. in appendix A.1 - some facts about equation 4, missing citation in definition 1.2 amongst others. I went over the proof of the main result and this appears to be correct. Furthermore, I find the connections to other learning rates (such as the cosine learning rate) to be rather hard to understand/interpret, in the current shape of the paper.



**Experience Assessment:**

I have published one or two papers in this area.

**Review Assessment: Checking Correctness Of Derivations And Theory:**

I assessed the sensibility of the derivations and theory.

**Review Assessment: Checking Correctness Of Experiments:**

I assessed the sensibility of the experiments.

**Review Assessment: Thoroughness In Paper Reading:**

I read the paper at least twice and used my best judgement in assessing the paper.

---

> ### Author Response · Authors · 2019-11-11
> **Response**
>
> Thanks for your thoughtful and valuable reviews!
>
> -We’ve improved the writing and fixed the typos in the paper. Regarding your suggestion about convergence analysis for routine smooth convex stochastic optimization problem, we do know how to get such a bound for the case of SGD + momentum + constant LR, which could be seen as a generalization of the convergence result in [Arora, Li, Lyu’2019] where momentum is turned off. The main idea is the same, because of the norm of the weight monotone increases (as shown in this paper), we can use the objective as a potential to bound the sum of the squared norm of gradients along the trajectory. Here the assumption is that the smoothness of scale invariant loss L(w) needs to be bounded outside the unit ball.  Due to the space limit, we didn’t put this result in the paper. But given your suggestion, we will try to put this result in the appendix of the next revision if time permits.
>
> However, it seems to be impossible to get a convergence rate for the case of SGD + momentum + constant lr + weight decay, as we’ve already given a non-convergence result for the toy example. If we further allowing step decay LR schedule, then when LR is sufficiently small, the dynamics is like gradient flow and it converges to stationary points of course.
>
> -The main point about the connection to other LR schedules is that we can get rid off WD by adopting the exponential version of the original schedule, which could accelerate the hyperparameter search.

---

### Official Review · AnonReviewer3 · 2019-10-24
**Official Blind Review #3**

**Rating:** 6

**Review:**

This exciting and insightful paper presents theorems (and illustrating examples and experiments) describing an equivalence of commonly used learning rate schedules and weight decay settings with an exponentially increasing learning rate schedule and no weight decay, for neural networks with scale-invariant weights. Hence, the results apply to a large set of commonly employed settings. The paper contains an interesting example of a neural network for which gradient descent converges if with batch normalization as well as with L2 regularization, but not when both are used.

From a theory viewpoint, the paper offers new insights into Batch normalization and other normalization schemes. From a practical viewpoint, the paper suggests a way to speed up hyper-parameter search, effectively allowing to consider learning rate and weight decay as one parameter.

A small gripe: this paper is a bit rough around the edges, and reads a bit like a draft (see comments on details below).


Detailed Comments / advice / questions
==================================

- It often takes a bit of searching to figure out what proof goes with what theorem / fact. I recommend to add to each occurrence of “Proof.” (before a proof) with a reference to the theorem or fact that is being proved, e.g. “Proof of Theorem 2.6”.

- The authors state that theorem B2 applies to “general deep nets”. In this theorem, the limit R_\infty could very well be zero (e.g. for networks with weights that are not scale-invariant), in which case the statement contains a division by zero. I wonder if the authors overlooked this or forgot to state an assumption in the theorem. Maybe I am missing something. Since this theorem does not appear to be critical to the main contributions of the paper, it may be easiest to remove the theorem if the division by zero is indeed a problem.


- On page 6, if c_w(x) is independent of w, would c(x) be more suitable?

- At the bottom of page 6, the authors state that “As a result, for any fixed learning rate, ||w_{t+1}|| ^4= …”. It appears that the authors get the expression for  ||w_{t+1}||^4 from the expression for ||w_{t+1}||^2 above (maybe by multiplying by ||w_{t+1}||^2?), but I don’t see how. Could the authors explain this? Maybe authors mixed up w_t and w_{t+1} by accident?

- The authors claim to “constrict better Exponential learning rate schedules”. Since the authors only perform a limited evaluation of their proposed learning rate schedule on CIFAR10, I suggest qualifying this statement.

- Theorem 1.1 does not introduce what \tilde{\gamma} is. It’s somewhat obvious, but I would state it nonetheless.

- The authors state that “…the exponent trumps the effect of initial lr very fast, which serves as another explanation of the standard wisdom that initial lr is unimportant when training with BN”. I don’t think that this constitutes a full explanation without further argument. I am also wondering if “another” is appropriate here: I am not aware of any (other) mathematically precise explanations of why initial learning rates do not matter in this set-up. If the authors are, they should cite it.

- The authors often forgot spaces before \cite commands. If you are trying to avoid a line break before the \cite command, you can use a tilde ~ , like this “Group Normalization~\cite{somepaper}”.

- Please introduce the abbreviation LR for “Learning Rate”, and always use the all-upper-case version (not “lr”).

- Definition 1.2 has a broken \cite .

- Theorem 2.7 should introduce \hat{eta}_t, but it doesn’t.

- Appendix A.1 contains a broken sentence (“as a function of…”)

- It’s odd that the proof for theorems B1 and B2 appear before theorems B1 and B2. I would restructure the appendices to improve this.

- Theorem B2 contains a stray “when”, and “exists” should be “exist”

- What the authors call “Proof of Theorem 2.4” in the appendix is really a proof of the rigorous version of Theorem 2.4 - Theorem 2.7! The proof should in my opinion be labeled “Proof of Theorem 2.7”

- The typo “eventsequation” should be replaced with something like “the events from equations”

- Replace the colloquialism “nets” with “networks”.

- Replace “BatchNorm” with “Batch Normalization”

- “COvariate” has casing issues

- “Riemmanian” should be “Riemannian”

- “BNhas” should be “BN has”

- The paper’s title states that the results are for batch-normalized networks, while the analysis appears to be more generally for networks with scale-invariant weights, which as the authors point out can arise from mechanisms other than batch normalization. Have the authors considered changing the paper’s title to better capture what their work applies to? In terms of discoverability, the authors would do the community a service by titling the paper in such a way that it captures the set-up well.

**Experience Assessment:**

I have read many papers in this area.

**Review Assessment: Checking Correctness Of Derivations And Theory:**

I carefully checked the derivations and theory.

**Review Assessment: Checking Correctness Of Experiments:**

I assessed the sensibility of the experiments.

**Review Assessment: Thoroughness In Paper Reading:**

I read the paper thoroughly.

---

> ### Author Response · Authors · 2019-11-11
> **Response**
>
> Thanks for your detailed and thoughtful reviews as well as the helpful suggestions!
>
> 1.  Our new revision tries to improve the writing and fixed the typos you have mentioned. We specified which theorem/lemma each proof corresponds to as you suggested. We also reorganized the sections in appendix.
>
> 2. You’re correct on that $R_\infty$ could be 0, thus making the statement meaningless. When $R_\infty= 0$, $D_\infty$ is also equal to 0 since $D_\infty <= 4* R_\infty$. A quick fix is just to replace the original statement by $D_\infty = \frac{2\eta\lambda}{1+\gamma}R_\infty$.
>
> 3. Random variable $c_w$ is actually a function mapping input data $x$ to a real number. This function indeed depends on $w$, but for different $w$, the random variable $c_w$ follows the same law(distribution).
> To avoid confusion, we modified this argument in the new revision and now we used the fact for each w, with constant probability, $\|\nabla_w L\|> \frac{C}{\|w\|}$, where $C$ is some other constant.
>
> 4. We indeed missed some subscription in the sentence “As a result, for any fixed learning rate, $\|w_{t+1}\|^4=$ …”.  The reason that $\|w_{t+1}\|^4>= \|w_t\|^4+2\eta^2c_w^2$ holds is that we can expand $\|w_{t+1}\|^4$ as$ (\|w_{t+1}\|^2)^2$, which is larger than $(\|w_t\|^2+\eta^2*c_w^2/\|w_t\|^2)^2 = \|w_t\|^4+2\eta^2c_w^2 + \eta^4*c_w^4/\|w_t\|^4$.
>
> 5. We’re sorry for the typo in the statement “…the exponent trumps the effect of initial LR very fast, which serves as another explanation of the standard wisdom that initial LR is unimportant when training with BN”. It should be that the initial scale of initialization is unimportant, and we give further explanation for this in the new revision. We don’t have a theory for this but we verified this intuition in experiments.
>
> Thank you so much for pointing this out!! And actually, the initial LR does matter a lot in practice.
>
> 6. As you suggested, we changed the title of the paper into “An Exponential Learning Rate Schedule for Deep Learning”.
>
> Thanks again for your helpful and thoughtful suggestions!! Given the new revision addressing all your concerns, would you consider raising your rating?

---

### Official Review · AnonReviewer2 · 2019-11-07
**Official Blind Review #2**

**Rating:** 8

**Review:**

This work makes an interesting observation that it is possible to use exponentially growing learning rate schedule when training with neural networks with batch normalization. This paper provides both theoretical insights and empirical demonstration of this remarkable property. In detail, the authors prove that for stochastic gradient descent (SGD) with momentum, this exponential learning rate schedule is equivalent to constant learning rate + weight decay, for any scale invariant networks, including networks with Batch Normalization and other normalization methods. This paper also contains an interesting toy example where  gd converges when normalization or weight decay is used alone while not when normalization and weight decay are used together.

Pros:

1. This paper gives new and important insight to the complex interplay between the tricks of network training, such as weight decay, normalization and momentum. The assumption and derivation are simple but the result is quite surprising. In classical optimization framework, it is common to keep the learning rate smaller than the 1/smoothness such that gd decreases the loss. However, the connection between exponential learning rate schedule and weight decay in common practice built by this paper suggests that the current neural net training recipe may be inherently non-smooth.

2. The experiment of this paper also suggests that in practice (with normalization layer), learning rate and weight decay coefficient can be packed into a single parameter, which reduces the effort needed for hyper-parameter tuning.

Cons:

1. Though it's obvious for the feedforward networks with normalization layers to be scale invariant, it's not the case for ResNet ( and the authors use this for experiment). And this needs to be clarified.
2. The writing of the proofs should be imporved.

Typos:

1. Definition 1.2 broke citation
2. Equation (1)  \eta_t should be \eta_{t-1}
3. Some facts about Equation 4, incomplete sentence
4 In thm B.2,R_\infty might be 0. So the authors can just delete the last equation on page 12 and use the equation above as the statement of the lemma.
5. In the first line of Equation (13), the appearance of ( \beta * e^{1-\beta} )^{ k/2 } seems to be a mistake


**Experience Assessment:**

I have published one or two papers in this area.

**Review Assessment: Checking Correctness Of Derivations And Theory:**

I assessed the sensibility of the derivations and theory.

**Review Assessment: Checking Correctness Of Experiments:**

I assessed the sensibility of the experiments.

**Review Assessment: Thoroughness In Paper Reading:**

I read the paper at least twice and used my best judgement in assessing the paper.

---

> ### Author Response · Authors · 2019-11-11
> **Response**
>
> We thank the reviewer for the careful reviews.
>
> In the new revision, we added a new section in appendix explaining the scale invariance in the modern network architectures. We also improved the writing of the proofs and fixed the typos.
>
> Thanks again for your appreciation.

---

### Official Review · AnonReviewer3 · 2019-11-21
**Official Blind Review #3**

**Rating:** 8

**Review:**

UPDATE TO MY EARLIER REVIEW
===========================

The authors improved the writing of the paper substantially relative to the first version they submitted, and fixed minor issues. I am changing my rating from "Weak accept" to "Strong accept". It is now a must-read paper for anyone doing research on deep learning.



MY EARLIER REVIEW
=================

This exciting and insightful paper presents theorems (and illustrating examples and experiments) describing an equivalence of commonly used learning rate schedules and weight decay settings with an exponentially increasing learning rate schedule and no weight decay, for neural networks with scale-invariant weights. Hence, the results apply to a large set of commonly employed settings. The paper contains an interesting example of a neural network for which gradient descent converges if with batch normalization as well as with L2 regularization, but not when both are used.

From a theory viewpoint, the paper offers new insights into Batch normalization and other normalization schemes. From a practical viewpoint, the paper suggests a way to speed up hyper-parameter search, effectively allowing to consider learning rate and weight decay as one parameter.

A small gripe: this paper is a bit rough around the edges, and reads a bit like a draft (see comments on details below).


Detailed Comments / advice / questions
==================================

- It often takes a bit of searching to figure out what proof goes with what theorem / fact. I recommend to add to each occurrence of “Proof.” (before a proof) with a reference to the theorem or fact that is being proved, e.g. “Proof of Theorem 2.6”.

- The authors state that theorem B2 applies to “general deep nets”. In this theorem, the limit R_\infty could very well be zero (e.g. for networks with weights that are not scale-invariant), in which case the statement contains a division by zero. I wonder if the authors overlooked this or forgot to state an assumption in the theorem. Maybe I am missing something. Since this theorem does not appear to be critical to the main contributions of the paper, it may be easiest to remove the theorem if the division by zero is indeed a problem.


- On page 6, if c_w(x) is independent of w, would c(x) be more suitable?

- At the bottom of page 6, the authors state that “As a result, for any fixed learning rate, ||w_{t+1}|| ^4= …”. It appears that the authors get the expression for  ||w_{t+1}||^4 from the expression for ||w_{t+1}||^2 above (maybe by multiplying by ||w_{t+1}||^2?), but I don’t see how. Could the authors explain this? Maybe authors mixed up w_t and w_{t+1} by accident?

- The authors claim to “constrict better Exponential learning rate schedules”. Since the authors only perform a limited evaluation of their proposed learning rate schedule on CIFAR10, I suggest qualifying this statement.

- Theorem 1.1 does not introduce what \tilde{\gamma} is. It’s somewhat obvious, but I would state it nonetheless.

- The authors state that “…the exponent trumps the effect of initial lr very fast, which serves as another explanation of the standard wisdom that initial lr is unimportant when training with BN”. I don’t think that this constitutes a full explanation without further argument. I am also wondering if “another” is appropriate here: I am not aware of any (other) mathematically precise explanations of why initial learning rates do not matter in this set-up. If the authors are, they should cite it.

- The authors often forgot spaces before \cite commands. If you are trying to avoid a line break before the \cite command, you can use a tilde ~ , like this “Group Normalization~\cite{somepaper}”.

- Please introduce the abbreviation LR for “Learning Rate”, and always use the all-upper-case version (not “lr”).

- Definition 1.2 has a broken \cite .

- Theorem 2.7 should introduce \hat{eta}_t, but it doesn’t.

- Appendix A.1 contains a broken sentence (“as a function of…”)

- It’s odd that the proof for theorems B1 and B2 appear before theorems B1 and B2. I would restructure the appendices to improve this.

- Theorem B2 contains a stray “when”, and “exists” should be “exist”

- What the authors call “Proof of Theorem 2.4” in the appendix is really a proof of the rigorous version of Theorem 2.4 - Theorem 2.7! The proof should in my opinion be labeled “Proof of Theorem 2.7”

- The typo “eventsequation” should be replaced with something like “the events from equations”

- Replace the colloquialism “nets” with “networks”.

- Replace “BatchNorm” with “Batch Normalization”

- “COvariate” has casing issues

- “Riemmanian” should be “Riemannian”

- “BNhas” should be “BN has”

- The paper’s title states that the results are for batch-normalized networks, while the analysis appears to be more generally for networks with scale-invariant weights, which as the authors point out can arise from mechanisms other than batch normalization. Have the authors considered changing the paper’s title to better capture what their work applies to? In terms of discoverability, the authors would do the community a service by titling the paper in such a way that it captures the set-up well.

**Experience Assessment:**

I have published one or two papers in this area.

**Review Assessment: Checking Correctness Of Derivations And Theory:**

I carefully checked the derivations and theory.

**Review Assessment: Checking Correctness Of Experiments:**

I assessed the sensibility of the experiments.

**Review Assessment: Thoroughness In Paper Reading:**

I read the paper thoroughly.

---

### Public Comment · ~Micah_Goldblum1 · 2019-11-08
**An Interesting Connection**

Hi Authors,
Thank you for your interesting paper.  I noticed that your work concerning learning rates for neural networks with batch normalization is related to our paper, which shows that an alternative to weight decay, which may stabilize effective learning rate, can improve performance for networks, especially those with batch norm.[1]  Please consider mentioning the relationship with our work in your next version.

[1] https://arxiv.org/abs/1910.00359

---

### Author Response · Authors · 2019-11-11
**A new revision is uploaded**

Besides improving writing and fixing the typos in the paper, we made the following major changes in the new revision:

1. We changed the title to “An Exponential Learning Rate Schedule for Deep Learning” for better discoverability as suggested by Reviewer #3.

2. We added a warm-up subsection in Section 2, proving the equivalence between exp LR and weight decay in the momentum-free case.

3. We provided a new perspective for understanding the almost equivalence between Step Decay + WD and our TEXP LR schedule, which says that with additional one-time momentum correction per phase, TEXP is **exactly** equivalent to Step Decay+WD. This explanation is conceptually easier than the original one(convergence of the LR growth rate) and thus we put this explanation in the main paper and defer the original one into the appendix.

4. We reorganized the experiment section and improved the writing.

5. In response to the reviews of Reviewer#2, we added a new section in the appendix named “Scale Invariance in Modern Network Architectures” explaining why various network architectures are scale invariant, including feedforward CNN and FC nets, ResNet, PreResNet. In detail, we gave an efficient algorithm for checking the sufficient condition of scale invariance and demonstrate how we could apply it onto the aforementioned architectures. The scale invariance of some architectures, like ResNet and PreResNet, is not entirely trivial to show.

6. We add a section named in appendix named “Viewing EXP LR Via Canonical Optimization Framework”, aiming at explaining why the efficacy of exponential LR in deep learning is mysterious to us, at least as viewed in the canonical framework of optimization theory.

---

### Decision · Program_Chairs · 2019-12-19

**Decision:**

Accept (Spotlight)

**Comment:**

After the revision, the reviewers agree on acceptance of this paper.    Let's do it.